# Endurance in Long-Distance Swimming and the Use of Nutritional Aids

**DOI:** 10.3390/nu16223949

**Published:** 2024-11-19

**Authors:** Álvaro Miguel-Ortega, Julio Calleja-González, Juan Mielgo-Ayuso

**Affiliations:** 1Faculty of Education, Alfonso X “The Wise” University (UAX), 28691 Madrid, Spain; 2International Doctoral School, University of Murcia (UM), 30003 Murcia, Spain; 3Physical Education and Sport Department, Faculty of Education and Sport, University of the Basque Country (UPV/EHU), 01007 Vitoria-Gasteiz, Spain; julio.calleja.gonzalez@gmail.com; 4Faculty of Kinesiology, University of Zagreb, 10110 Zagreb, Croatia; 5Faculty of Health Sciences, University of Burgos (UBU), 09001 Burgos, Spain; jfmielgo@ubu.es

**Keywords:** swimming, endurance, energy demands, physiological demands, nutrition, body composition, ergogenic aids

## Abstract

Background: Long-distance swimmers exert energetic, physiological, and neuromuscular demands that must be matched with adequate body composition to improve their performance in long-distance swimming. Objectives: This review aims to compile all available information on energetic and physiological demands, optimal body composition, nutrition, and ergogenic supplements in long-distance swimming. This will provide an understanding of the specific challenges and needs of this sport and will help swimmers and coaches design more effective training and nutrition plans to optimise performance and achieve their goals. Methods: Databases such as Web of Science, SciELO Citation Index, MEDLINE (PubMed), Current Contents Connect, KCI-Korean Journal Database, and Scopus were searched for publications in English using keywords such as swimming, endurance, energy demands, physiological demands, nutrition, body composition, and ergogenic aids, individually or in combination. Results: There is convincing evidence that several physical indicators, such as propulsive surface area, technical, such as stroke rate, and functional, such as hydration strategies, are related to swimming performance and body composition. Each athlete may have a specific optimal body fat level that is associated with improved sporting performance. The nutritional needs of open water swimmers during competition are quite different from those of pool swimmers. Conclusions: Swimmers with an adequate physique have a high body muscle mass and moderately related anaerobic strength both on land and in the water. These general and specific strength capacities, which are given by certain anthropometric and physiological characteristics, are seen throughout the work, as well as ergogenic and nutritional strategies, which have an important impact on long-distance swimming performance.

## 1. Introduction

Long-distance swimming places exceptional physiological and/or psychological demands on swimmers [1], both during competitions and training [2]. These demands are dependent on energetic factors that condition aerobic endurance, and are also influenced by physiological factors such as muscular strength or anaerobic power, anthropometric factors such as height or body composition, nutritional factors [3], or nervous factors such as neuromuscular coordination [4], all of which are described in the scientific literature [5,6]. Special mention should also be made of the ergonomic support for long-distance swimming [7].

From a physiological point of view, in swimming, among all the performance factors mentioned above, the aerobic system is the energy system that is most in demand in this type of event [7], although the anaerobic lactic energy system is also activated, depending on the type of event. However, in prolonged cyclic events such as this one, the contribution of this system decreases [8] and is only activated at times of high demand. Therefore, performance improvements must come from the development of the aerobic and, to a lesser extent, the anaerobic energy system [9].

Similarly, maximal oxygen uptake (VO_2max_) in swimming is affected by external conditions that create different metabolic and biomechanical scenarios that affect energy demands [10]. In addition, factors such as horizontal body position [10,11], which increases pressure and reduces blood flow and muscle perfusion [12], as well as having less muscle mass in the upper body, which has more fast twitch fibres than the lower body [13], can slow down the VO2 response [10] and influence energy demands. However, all energy systems contribute in some way to exercise intensity. Each system is best adapted to different stimuli [14]. Therefore, all energy systems are important in determining performance. It has also been observed that swimming speed and distance determine these intensities. Long-distance swimmers are used to swimming at low to moderate speeds [15,16], maintaining a high volume and low to moderate intensity. This is related to specific anthropometric parameters.

Open water swimmers tend to be shorter and lighter, with less lean muscle mass than pool swimmers [15,17,18]. This may be because less absolute power is needed to complete open water events compared to shorter distances [19]. It may also be because less skilled swimmers have traditionally competed in open water events.

Nutritional recommendations for training and competing in open water are based on recommendations for pool swimming or extrapolated from other sports with similar physiological requirements [19]. In this type of competition, nutrition should focus on optimising hydration and maintaining glycogen stores [19]. During the event, swimmers rely on their own sources of fuel and fluids, only using feed zones when tactically appropriate. As the duration of the race is extended, feeding zones during long-distance swimming become more important to reduce the stress associated with prolonged environmental exposure, as it is important that the amount of calories supplied to the athlete during intense training is sufficient to avoid a relative energy deficiency [20]. Although depending on the duration of the test, a significant reduction in subcutaneous adipose tissue may be observed throughout the test [21,22]. However, the literature on this type of test and its relationship to energy expenditure is not extensive [23].

On the other hand, the authors have studied nutritional supplements that can improve performance in long-distance swimming [24]. Some products can increase strength and help reduce metabolic acidosis [25], which is of particular interest to open water swimmers. Caffeine is beneficial for aerobic activities [26,27] and creatine for anaerobic sprints [28,29]. However, ephedrine and pseudoephedrine have detrimental effects on the cardiovascular system [30]; β-hydroxy-β-methyl butyrate may be useful for untrained individuals [31,32], while pyruvate does not improve performance [33].

After reviewing all these aspects, the authors have not found any comprehensive review that analyses all the factors that can influence performance in long-course swimming events. Therefore, assessing how each of these elements affects performance in this type of event is crucial to better understand the phenomenon and to develop more effective training and preparation strategies. Unfortunately, the available scientific literature lacks a comprehensive approach that integrates all these elements in a systematic and rigorous way. This difficulty represents an opportunity for future review and meta-analysis studies to address this need and provide a more complete picture of the determinants of performance in long-course events.

Thus, our aim is to compile all available information on energy requirements, physiological requirements, body composition, nutrition, and supplementation in long-distance swimming. This will help us to understand the challenges and needs of this sport. It will also allow us to discover aspects that can help swimmers and coaches create effective training and nutrition plans to improve performance and achieve their goals.

## 2. Materials and Methods

### 2.1. Sources of Information

This article is a narrative review focusing on considerations related to open water performance. The review was conducted following the Preferred Reporting Items for Systematic Review and Meta-Analyses (PRISMA) guidelines [34].

The PICOS model was used to determine the inclusion criteria referred to in [35] (Figure 1).

The studies included in this review had to meet certain criteria. (I) The study population had to be swimmers. The articles had to examine energy needs, physiological needs, body composition, nutrition, and ergogenesis in swimming (II). Study designs had to be non-randomised (III). Relevant studies were selected for the review (IV). Exclusion criteria were applied to experimental research protocols. (i) Studies with participants from other disciplines or with previous disorders were excluded (ÁM-O). (ii) Articles on other sport populations, abstracts, non-peer-reviewed articles, and book chapters were also excluded (ÁM-O). (iii) Finally, studies unrelated to energy demands, physiological demands, body composition, nutrition, or ergogenic aids were excluded (ÁM-O).

Author ÁM-O conducted structured electronic searches of the scientific literature in various databases, such as Web of Science (WOS), SciELO Citation Index, MEDLINE (PubMed), Current Contents Connect, KCI-Korean Journal Database, and Scopus. The search included a combination of the following keywords: ((‘swimming’ [MeSH Terms] OR ‘swimming’ [All Fields]) OR (‘ultra endurance’ [MeSH Terms] OR ‘ultra endurance’ [All Fields]) AND (‘energy’ [MeSH Terms] OR ‘energy’ [All Fields]) AND (‘physiology’ [MeSH Terms] OR ‘physiology’ [All Fields]) OR (‘body composition’ [MeSH Terms] OR ‘body composition’ [All Fields]) OR (‘nutrition’ [All Fields] OR ‘nutrition’ [All Fields])) AND (‘ergogenics’ [MeSH Terms] OR ‘ergogenic’ [All Fields]). These keywords were chosen based on the opinion of the authors (ÁM-O, JM-A and JC-G), the scientific literature review, and controlled vocabulary such as Medical Subject Headings (MeSH). The search was not restricted by publication date but was filtered to include only English language studies in humans. The search was not restricted by publication date but was filtered to retrieve only human studies written in English. The last search was conducted on 4 September 2024. To analyse and discuss the most relevant articles in the field, we also examined the articles from the reference lists of the articles retrieved in the search (snowball strategy [36]).

### 2.2. Study Selection

Publication titles and abstracts were identified using the search strategy described above and cross-referenced to identify duplicates (ÁM-O and JC-G). All trials assessed for eligibility and classified as relevant (ÁM-O and JC-G) were retrieved. In addition, the reference section of all relevant articles was examined [37]. From the information contained in the full articles, inclusion and exclusion criteria were used to select eligible trials for insertion into this systematic review. There was no disagreement between AM-O and JC-G regarding the eligibility of studies.

### 2.3. Data Extraction

After applying the inclusion/exclusion criteria to each study, the following data were extracted (ÁM-O): study source (author/authors and year of publication); sample population, indicating the number of participants; methods; intervention characteristics; and significant differences between study groups. After data extraction, the included studies were grouped according to the following criteria: (1) energy demands; (2) physiological demands; and (3) the relationship between these demands and QC and nutrition, as well as performance aids. To minimise errors, data extraction and group formation were discussed among the authors until a final consensus was reached (ÁM-O, JM-A, and JC-G).

It is important to note that the process of data extraction and clustering was not a straightforward task as some studies lacked certain information or had inconsistencies in their reporting. However, by carefully reviewing the studies and discussing discrepancies between authors, a comprehensive understanding of the relevant data was achieved.

Furthermore, it is worth mentioning that the studies included in this analysis were selected based on rigorous criteria, such as the use of randomised controlled trials or the inclusion of a control group. This ensured that the studies were of high quality and provided reliable information for the analysis.

Overall, the process of data extraction and pooling played a crucial role in the analysis of the studies as it allowed for a systematic and comprehensive examination of relevant data. Through this process, important patterns and trends were identified, providing valuable insights into the relationship between energy demands, QC, nutrition, and performance.

### 2.4. Assessing the Quality of Experiments: Risk of Bias and Levels of Evidence

To carefully consider the possible limitations of the included studies to obtain reliable conclusions, following the Cochrane Collaboration Guidelines [38], the statement Strengthening the Reporting of Observational Studies in Epidemiology (STROBE) guidelines for reporting observational studies [39] (Figure 2) was used to assess the quality of the publications.

Two authors (AM-O and JC-G) independently assessed methodological quality and risk of bias with any disagreement resolved by assessment by a third party (JM-A), with the Cochrane Collaboration Guidelines [38]. The following scale was used to classify study quality: (1) good quality (>14 points, low risk of major or minor bias); (2) fair quality (7–4 points, moderate risk of major bias); and (3) poor quality (<7 points, high risk of major bias) [39].

The checklist items were classified into different domains: random sequence generation (selection bias), allocation concealment (selection bias), blinding of participants and staff (implementation bias), blinding of outcome assessment (detection bias), incomplete outcome data (attrition bias), selective reporting (reporting bias), and other types of bias. They were classified as ‘low’ if they met the criteria for insignificant risk of bias (plausible bias unlikely to seriously alter the results) or ‘high’ if they met the criteria for elevated risk of bias (plausible bias that seriously undermines confidence in the results). If the risk of bias was unknown, it was considered ‘unclear’ (plausible bias that raises some doubt about the results) [38].

Most of the trials assessed showed a lack of clarity in the criteria for other types of bias due to incomplete reporting or omission of relevant variables, such as QC or nutritional patterns. The robustness and real-world applicability of the study’s ergo-nutritional aids were assessed using the tool developed to evaluate performance nutrition research [40] (Figure 2 and Table 1).

The level of evidence of the chosen studies was determined using the Oxford quality scoring system (Table 2), a procedure widely used worldwide [46,47]. This tool for assessing the quality of clinical studies has become an international standard of reference [48], allowing the methodological quality of trials to be assessed in an objective and standardised manner, which in turn facilitates the interpretation and comparison of the results obtained. This instrument is especially relevant in the context of systematic reviews and meta-analyses, where the assessment of the quality of the included studies is essential for the validity of the conclusions.

## 3. Results

An initial search of the scientific literature identified thirty-one articles related to endurance swimming and other keywords for inclusion in this systematic review. Of these, twelve articles unrelated to BC, energy demands, or physiological demands were excluded. We also eliminated five that dealt with other disciplines (did not meet the inclusion criteria) (Figure 3). Table 3 shows the summary of the included studies.

The energy requirements of endurance-trained swimmers are shown in Table 4. Also, about this aspect, Table 5 shows the analysis of energy balance in swimmers based on the information on caloric intake provided by the swimmers themselves and the calculations of energy expenditure. However, if we focus on the consumption of a macronutrient such as protein, this can be seen in Table 6.

As for FM (fat mass), there is compelling evidence in several studies that have analysed different physical, technical, and functional indicators related to swimming performance and their relationship with FM as can be seen in Table 7. Although there is no reference data on the ideal body composition of swimmers for each sport discipline, each athlete may have an ideal level of body fat that is related to higher sports performance and better results, as can be seen in Table 8. This is related to the nutritional needs of open water swimmers during competition, which are hugely different from those of pool swimmers. The key considerations for the main types of races in this aspect can be summarised in Table 9.

## 4. Discussion

As can be seen, our study aims to bring together all available information on energy and physiological demands, BC, nutrition, and ergogenic aids in long-distance swimming to understand the demands and requirements of long-distance swimming and what factors can help swimmers and coaches develop effective training and nutrition plans to improve performance and achieve their goals.

### 4.1. Energy Requirements

Swimming athletes must have characteristics such as power, speed, and endurance, attributed to the energy systems of phosphates, lactic acid, carbohydrates (CHO), fats, and proteins. However, understanding the specific physiological needs is difficult, as the literature on long-distance swimming and energy expenditure is relatively limited [18,23].

The energy contribution in swimming is a large structure, as 44% of the energy contribution [106]. The most used method to estimate this contribution is the oxygen deficit, written to estimate the anaerobic capacity [50] during maximal individual and advanced swimming techniques. This method has been used to estimate the energy contribution during maximal individual effort combined with advanced swimming techniques [102,107,108], although external conditions can create different scenarios at the metabolic and biomechanical level in swimming, and thus affect the normal VO_2_ response [10].

All three energy systems contribute to all exercise intensities, but each may be best suited to supply energy depending on the stimulus [109]. Thus, all energy systems significantly impact performance, especially in high-intensity exercise. For this reason, creating a linear relationship between VO_2_ and swimming intensity is the main problem when estimating oxygen demand, with higher-level swimmers contributing more aerobically than moderate-level swimmers [110] (Table 4).

Thus, the changing energy needs of a swimmer are reflected in the volume of training/competition and the growth or goals of modifying their physique. It has been estimated that male swimmers need 15–20 MJ/day (3600–4800 Kcal/day) and female swimmers 8–11 MJ/day (1900–2600 Kcal/day). Examples of this aspect can be seen in Table 5 [54,56,58]. However, other studies show large apparent mismatches between energy intake and energy expenditure (up to 10 MJ/day or 2400 kcal/day) when training volumes and intensities are large [57]. We must also take into consideration that the ability to adjust energy needs to expenditure during swimming seems to be different for both sexes [111,112,113]. Thus, during training, male swimmers can increase their energy intake, mainly by increasing their CHO intake to meet the increased fuel needs [114], although this increase does not always happen [115]. In contrast, female swimmers are less able to make these changes instinctively, a fact that may be due to lower levels of lean muscle mass [55], higher swimming efficiency [116,117], or underreporting of dietary intake, especially about to BC.

However, on the other hand, swimming has been anecdotally associated with increased appetite and an increased risk of overeating compared to other sporting activities. One potential reason for this is that heat dissipation in chilly water reduces thermoregulatory stress. After swimming sessions (≤70% VO_2max_) [118], no differences in appetite and ad libitum intake have been found. Exercise in cold water (20 °C) consumes 40% more energy after exercise than exercise in thermoneutral or control conditions [119]. But whether the increased food intake after exercise in water is due to core temperature regulation or some other mechanism remains to be discovered, and further research is needed in this aspect.

Regarding CHO requirements in swimming, it has been studied that swimming training can deplete muscle glycogen. Completion of high-volume training blocks depends on adequate CHO intake [115].

One study [115] found that swimmers who did not increase their CHO intake in response to a sudden increase in training volume experienced more fatigue, muscle soreness, and difficulty completing designated workouts compared to swimmers who spontaneously increased their CHO intake (8.2 g/kg body mass, or BM, per day) to maintain muscle glycogen stores. This does not imply that the baseline training diet necessarily presents a high amount of CHO. CHO intake should allow for muscle glycogen recovery when high-quality, high-intensity training is required. In fact, another study showed that swimmers with a moderate CHO load of 6 g/kg bw/day met glycogen requirements but gained no performance advantage from consuming a higher amount of BM (12 g/kg/day) [120]. To enable high-intensity training, daily CHO intake should be adjusted to accommodate the changing fuel requirements of the training program.

Now, if we focus on protein intake specifically (Table 6), swimmers should follow the current guidelines of consuming 0.3 g/kg BM of protein to maximise the early protein synthesis response to the exercise stimulus.

Therefore, for swimming athletes to reach their maximum performance potential, they need to develop characteristics such as power, speed, and endurance. The phosphate energy system and the lactic acid energy system, as well as the aerobic combustion of CHO, fats, and proteins, provide energy for these physical elements. However, understanding the specific physiological needs of these athletes is difficult because swimming is performed in water.

As for metabolic power (the energy expended in a unit of time), in swimming, it increases exponentially as a function of speed (as a “measure” of exercise intensity). Thus, the swimmer who can use as much energy as possible while swimming at maximum speed will be able to achieve his or her highest metabolic power [45].

Thus, the energy demands of training are significant because the training of long-distance swimmers involves large volumes of work completed at high aerobic intensities (<75% of training [15]). High energy requirements (23 MJ/day) have been demonstrated in female swimmers performing large training volumes (17.5 km/day [57]), and male swimmers with similar training would have similar energy requirements (4–29 MJ/day [115]).

This type of event requires long hours of effort, which results in a decrease in average swim speed as the distance increases.

Long-distance swimmers rely on their body’s energy expenditure for prolonged periods. The best swimmers can effectively fuel their body’s engine, but their functional capacity can be compromised by improper stroke patterns.

Water resistance affects swimming efficiency. Long-distance swimmers tend to use more glycogen at slower speeds, as they have fewer fast twitch muscle fibres. They have more slow twitch fibres, which allows for a greater aerobic metabolism of glucose and fat. This allows them to swim for longer periods before depleting their muscle glycogen.

During long-distance events, there is a significant decrease in muscle concentrations of phosphocreatine (PCr) (75%), ATP (25%), and glycogen (34%). This change in substrate concentration occurs in the first 2–3 min of the test. In addition, the type of effort exerted in these tests is accompanied by a significant increase in blood and muscle lactate concentrations, although this increase is less than that observed in tests of shorter duration.

Even so, individual differences between athletes in energy demand mean that there may be significant differences in the energy expenditure of the same athlete depending on the day of the week, which is an argument in favour of controlling the energy balance of certain athletes [121]. It has also been observed that the sex of the athletes significantly differentiated the average energy expenditure of male and female athletes, which was higher in male athletes than in female athletes [122]. Thus, the actual energy needs of an athlete may differ significantly from the energy intake standards recommended in the literature [121]. Consequently, using the norms as the sole reference for developing dietary plans for athletes may lead to a miscalculation of caloric intake and disturbances in energy balance [123].

### 4.2. Physiological Demands

Since the beginning of the 20th century, extensive research has been conducted on the energetics of individual and cyclic sports [45]. Oxygen uptake (VO_2_) is one of the most studied variables in exercise physiology since then. Mainly due to the restrictions imposed by the aquatic environment, the number of studies with a direct assessment of VO_2_ and kinetics in swimming is significantly lower [124,125].

Thus, this type of test is swum at a relative intensity of 5–12% below VO_2max_. This intensity usually represents 110–115% of the individual anaerobic threshold.

The results of the studies carried out show that the fast and slow components of VO_2_ change progressively depending on the intensity and duration of the swimming events. During high intensity and short duration events, an immediate and continuous increase in VO_2_ is observed, highlighting the importance of the aerobic energy pathway even in short and very demanding swimming events.

In addition to confirming the relevance of the aerobic and anaerobic energy systems in short events [11,126,127], the results indicate that a shorter cardiac phase and higher VO_2_ levels are associated with better swimming performance. The absence of significant gender differences suggests that faster VO_2_ kinetics is a general adaptation to swimming training [128].

On the other hand, in long sets performed at low-moderate intensity, swimmers showed greater stability in all parameters of VO_2_ kinetics. However, at heavy and severe intensities, there was faster VO_2_ kinetics with a slow and pronounced VO_2_ component. This indicates that the physiological response of the organism varies according to the duration and intensity of the swimming test.

Furthermore, it has been observed that swimmers with more efficient VO_2_ kinetics, with a shorter cardiac phase, and higher VO_2_ levels tend to perform better in swimming events. This suggests that long-term training and adaptation may improve the efficiency of energy systems and thus swimming performance.

This confirms the importance of aerobic energy supply (with muscle glycogen as the main fuel source) in long-distance and long-distance swimming events [129,130], which depend on the duration and intensity of exercise as well as the swimmer’s training status [131,132]. However, as exercise intensity increases, not considering the anaerobic contribution may reduce energy expenditure [133].

Long-distance events are part of the long endurance specialities I [134], which indicates that 80% of the energy is provided by aerobic energy metabolism and 20% by anaerobic metabolism (lactic and alactic), while the stimulation of the oxygen transport system is maximal or near maximal.

Some arguments that are often made to confirm that this type of event is related to the oxygen transport system is that elite swimmers competing in these events have some of the highest oxygen consumption values within the swimming specialities. However, higher correlations have been found between anaerobic threshold and performance in this type of exercise than between VO_2max_ and performance. Therefore, it seems likely that performance in these tests is also related to the muscle’s ability to utilise CHO (limiting factor) and the availability of oxygen through both energy pathways. It has been observed that anaerobic capacity [135] is also a determinant of endurance tests, whereas economy measured at medium or low intensities is not [136]. For these reasons, it appears to be important to have a competent level of mixed aerobic and anaerobic strength endurance that allows the swimmer to maintain an effective stroke length to the end of the race while applying force and resisting local muscle fatigue.

Long-distance swimmers tend to have a natural inclination towards aerobic metabolism, which means they can swim for prolonged periods at fast speeds, but not at top speeds without experiencing elevated levels of acidosis. But as their VO_2max_ and anaerobic threshold are often higher, they may struggle to start the race fast enough to be competitive and are unlikely to have a sufficient speed for a fast finish. For these reasons, there is now a tendency among distance swimmers to excel at shorter distances [137].

There are significant differences in peak blood lactate values between competitive efforts in middle- and long-distance events. This confirms that the efforts are hugely different in terms of the involvement of various energy sources [138,139].

Long-distance endurance events lasting between 45 min and 3 h usually coincide with the depletion of muscle glycogen stores. Muscle glycogen utilisation will occur in type I and IIa muscle fibres from the start of exercise, whereas glycogen utilisation in type IIa and Ilb fibres is lower and begins to occur in the second third of the event. Glycogen stores take more than 24 h to replenish, even if food containing 50–60% CHO is eaten. If these intense exercises are repeated on consecutive days, there is a gradual depletion of muscle glycogen stores that may require several days of rest or light training to fully recover.

During these tests, the decrease in muscle PCr reserves and the increase in muscle lactate concentrations are much lower than those observed at other distances. It is important to note that it is common to perform a final sprint during the competition, which can stimulate anaerobic glycogenolysis and, as a result, increase lactate concentration.

During these events, actively participating muscles draw substantial amounts of glucose from the blood. To maintain blood glucose levels, 75–89% of the glycogen stored in the liver is released into the blood, along with 11–25% of other substrates such as lactate, glycerol, pyruvate, or alanine.

In contrast to the long-distance tests, there is a significant involvement of lipids in this type of test. The lipids used come both from the blood (circulating fatty acids, released by adipocytes) and from within the muscle.

Protein utilisation is small (5%). However, when muscle glycogen stores are low, muscle glycogen utilisation increases considerably and is accompanied by an increase in blood urea concentration at rest and during exercise. Monitoring urea levels during exercise and at rest the following day provides insight into the degree of muscle protein utilisation during physical activity.

In summary, the following factors will be key to a long-distance swimmer’s performance:Anaerobic threshold level, which is especially important to maintain a high percentage of VO_2max_ without accumulating lactate (80–91%), to maintain a high average speed with lactate concentrations between 3–5 mmol/L.Aerobic capacity (VO_2max_). The higher the VO_2max_, the easier it is to use oxygen in anaerobic threshold conditions.Satisfactory level of aerobic strength endurance to apply and maintain stroke power in aerobic conditions.Good ability to distribute effort tactically and execute it effectively and economically in open water (technique).Muscle and liver glycogen. Increasing these stores is of utmost importance to not affect performance over these distances. In addition, depending on the duration of the race, it is advisable to take CHO during the effort to maintain a high intensity.Fat mobilisation (aerobic threshold). Good capacity of fat metabolism to provide energy, which can reach up to 40%.Good ability to distribute effort tactically and execute it efficiently and economically (technique).

At the hormonal level, long-distance events partially inhibit anabolic hormones and decompensate metabolism. Cortisol levels increase, supplementing energy substrates through lipolysis, proteolysis, and gluconeogenesis, which can delay recovery.

The blood lactate concentration is maintained at levels like those at rest, sometimes even below 2 mmol/L [139]. However, during competition or a final sprint, it can reach values of 2–3 mmol/L [140].

The participation of lipids in energy production is incredibly significant, gradually increasing throughout the event. Most of the lipids used come from blood fatty acids, although muscle lipid reserves are also used to a significant extent. Skeletal muscle protein oxidation is also significant, providing between 4% and 8% of the total energy required for muscle contraction.

In this type of swimming event, as the relative exercise intensity is less than 60–65% of VO_2max_ and 80–85% of individual anaerobic threshold, aerobic metabolism accounts for almost 100% of energy production, derived from fatty acid oxidation (about 60%), blood-derived glucose (about 5%), muscle glycogen (about 35%), and protein oxidation (about 5%) [141].

The execution of movements lasting several hours is monotonous and is regulated at the spinal motor level. The cessation of effort is caused by overload phenomena in the motor system (muscle ramps, tendon pain, etc.). In this respect, it is crucial that the ligamentous and tendon tissues involved can withstand the effort.

The heart rate is low, ranging between 120 and 150 beats per minute. The systolic volume copes with the high stress, allowing a stable supply of oxygen to the myocardium. Maximal oxygen consumption is 50–60% when movements are of lower intensity. VO_2max_ values of these long-distance athletes are typically low (55 to 65 mL/kg/min) due to the low working speeds.

As we already know, VO_2_ is calculated from blood flow and the difference in oxygen between arterial and venous blood. This is directly related to the total energy expenditure [142]. Thus, VO_2_ is commonly used to indicate the aerobic intensity level of an individual or activity, allowing an objective comparison of intensity between different types of exercise [143].

In relation to this aspect, and in comparison, to similar land-based exercises, VO_2_ max and heart rate values were lower during deep and shallow water exercise. However, depending on water depth and the type of exercise, they may be higher during aquatic calisthenics or underwater treadmill exercises [144]. However, overall, the perception of exertion during deep water exercise was generally like that of land-based exercise during maximal exertion [145].

Therefore, performance in this type of test depends on high aerobic endurance, such as the following:High aerobic efficiency at low intensity effort levels below the anaerobic threshold (50–60 60% VO_2max_);Mobilisation of proteins and fats can contribute up to 65% of total aerobic energy;Anaerobic threshold to help maintain lactate concentrations between 2–3 mmol/L;Muscle and liver glycogen is extremely low, and CHO intake is desirable to maintain higher intensity during exertion;Aerobic strength: in anaerobic threshold conditions, increasing VO_2max_ will increase oxygen utilisation;Temperature control is necessary to administer fluids to avoid losses through perspiration (more than 3 litres) and electrolytes (Na+, Cl-, H+ and Mg) to regulate internal temperature and maintain nerve and muscle conduction functions;Tendon and ligament tissue strength; the ability of the locomotor system to withstand these stresses is particularly crucial.

### 4.3. Body Composition

The ability to generate propulsion while reducing resistance through water and a delicate balance between muscle mass and the appropriate morphology are essential for swimming performance [2], relying on muscle strength and power-generating properties. In turn, these characteristics are related to BC, especially lean muscle mass. Muscle mass is logically related to strength and propulsion and thus to swimming speed, although it has been documented [43] that anthropometry has no correlation in ultra-distance swimming. Furthermore, it has also been reported that anthropometric characteristics specific to open water ultra-distance swimmers were associated with race time in men, but not in women [42]. In contrast, anthropometric characteristics were not related to race performance in male swimmers in a pool-based ultra-endurance test [41]. In particular, the physical and metabolic characteristics of elite open water swimmers have been examined [17] and it was reported that these swimmers possessed aerobic metabolic alterations that translated into increased performance in long-distance swimming. Although in competitive ultra-endurance pool swimmers in a 12-h event, no anthropometric variables were related to the total distance achieved [41].

Competitive swimming usually experiences this interaction [76,97] unless the increase in muscle mass leads to an excessive increase in body size, which reduces leanness and thus increases shape [146]. Male and female swimmers have similar body structures, and even a small variation in their anthropometric measurements, such as height or muscle mass, can affect their performance [146]. Therefore, to increase your likelihood of success in competition, some anthropometric variables can help you choose the right distance to swim. Variation in swimming performance demonstrates the importance of body structure and muscle mass on performance. The literature [97] finds that upper body musculature is largely responsible for propulsion and force production in swimming [76].

Unlike running, where the oxygen cost is similar for both sexes, the energy cost of freestyle swimming is higher (i.e., lower economy) in men than in women [146]. The greater economy in women has been attributed to a smaller body size, resulting in lower body resistance, lower body density with a higher percentage of fat, and shorter lower limbs, resulting in better horizontal and aerodynamic positions [147,148]. This greater economy in women could be an advantage in ultra-swimming performance. Thus, the observed changes in skeletal muscle mass among male swimmers are like those of other researchers who have shown that excessive increases in endurance activity levels will lead to a reduction in muscle mass [22,149,150].

In a comprehensive study examining the body composition of a group of athletes [151], FFM was found to increase significantly throughout the competitive season. This suggests that the training programs and proper nutrition implemented during this period were effective in promoting the development of lean muscle mass in the participants.

Similarly, another study [76] revealed a marked decrease in total body mass and skinfold thickness, accompanied by an increase in FFM from preseason to approximately two weeks before the end of the season. These findings indicate that swimmers were able to optimise their body composition through the careful planning and execution of their training cycles and nutritional strategies throughout the year.

These relevant aspects of body composition changes throughout the sport season must be taken into crucial consideration by coaches and sport professionals when designing and implementing training and recovery plans for their athletes. Proper periodisation and an integrated approach to workload management, nutrition, and recovery can contribute significantly to the long-term performance and well-being of athletes.

Tracking changes in muscle mass over the course of a season can be extremely beneficial for athletes and coaches. These changes in muscle mass are often closely related to changes in physical and athletic performance.

Thus, by regularly monitoring the body composition and muscle mass of athletes, important trends and patterns can be detected. For example, a sustained increase in muscle mass often correlates with improvements in strength, power, and endurance. Similarly, a decrease in muscle mass may indicate fatigue, overtraining, or impending injury that can negatively affect performance.

This systematic monitoring allows coaches to adjust training programs, nutrition, and recovery more accurately and effectively. In this way, the athletes’ muscle development and athletic performance can be maximised throughout the season. In addition, the data collected can be valuable in assessing the effectiveness of training methods and making strategic adjustments when necessary.

For example, long-distance swimmers have been found to have a lower weight and lower percentage of lean muscle mass than swimmers of other distances, based on the anthropometric data collected [15,17,18]. This could be because less absolute power is required to successfully complete this type of event [126] with these swimmers having a very high oxygen consumption (80 mL/min/kg for men and 66 mL/min/kg for women, respectively) [15].

The morphological shape or muscle distribution of swimmers in relation to swimming performance is less studied. We found one study [15] in which the authors stated that the cross-sectional area of these muscles could be an indicator of propulsion in different types of events. Muscular architectural characteristics, such as muscle thickness and triceps brachii fascicle length, and swimming performance have also been significantly related [94]. Furthermore, together with body height, triceps brachii length was one of the best predictors of performance. These findings support other studies [152], in which propulsive force was calculated using an arm muscle cross-section [153,154]. Thus, at present, it is probably more important to achieve the corresponding body length through training and muscle mass distribution [76,155] by reducing skinfolds during the season.

Most research has focused on biomechanical and physiological assessment due to the assumption that BC plays an important role in sports [156], especially in swimming, where BC monitoring is a valuable tool for optimising competitive performance by monitoring the effectiveness of body adaptation to the training process [157]. Because this sport is quite demanding from a training point of view, most research has focused on a biomechanical and physiological assessment [158,159], as BC and strength have been considered as indicators of performance in this specific sport [160].

The BC of athletes in terms of fat, bone, muscle, and water mass is crucial in sport [88], especially in relation to the health and performance of athletes [156]. Furthermore, it has been shown that morphological characteristics of young athletes can affect their swimming and testing performance [152]; in particular, regional and whole-body FFM affects anaerobic reserves, mass, and short-term performance.

These anthropometric characteristics of aquatic athletes vary significantly between disciplines and genders. Swimmers tend to have a tall body and focus on muscular strength and power to improve propulsion. Most male swimmers have more muscle and less body fat than female swimmers. Male swimmers have twice as much lean mass within and between seasons as female swimmers [76].

For example, male long-distance swimmers are typically shorter and lighter than the typical average swimmer [89]. Female long-distance swimmers often have much thicker skinfolds than other swimmers [17]. So, throughout the swimming season, training should be associated with regular increases in lean tissue and regular decreases in fat mass [56,161,162,163].

Previous research found a body mass index (BMI) of 22.78 in swimmers [152]. An analysis of the 2012 Olympic Games [146] found that male and female swimmers had a mean BMI of 23, leading to the conclusion that power and VO_2max_ were the main causes of differences in race performance among elite athletes in these swimming events. The current BMI results [104] confirm that swimmers who train regularly have BMI values within the healthy range (18.5 to 24.9). However, these findings also indicate that BMI alone is not the best indicator or predictor of performance.

Thus, it has been observed that body shape resistance (frontal area) and skin friction decrease with lower BMI, but the contractile potential of the DC increases the propulsive force to swim faster [73], as lower BMI has been shown to translate into lower endurance [73]. It has also been shown in previous research that fat reduction increases muscular and cardiorespiratory endurance (CE), as well as speed and agility [101]. It appears that fat mass (FM) and FFM have an impact on swimmers’ performance [78,80]. Previously, the effects of competitive swimming on BC were investigated [127] and it was found that swim training did not affect bone density and FFM in the lower limbs (LL), although FM decreased. Despite this, it has been shown that the period of intense training during a competitive swimming season is associated with a significant increase in FFM and a decrease in FM [164].

Thus, muscle size is strongly related to strength and power [80,96], and athletes with larger muscles, or FFM, can perform specific movement efforts with more muscle strength, which increases speed, quickness, acceleration, and agility [75,82].

In general, sub-Olympic swimmers have been found to be less tall than competitive swimmers, especially elite Olympic swimmers. Current swimmers are taller, heavier, and larger than previous elite male swimmers, according to longitudinal data from Russian national swim teams [77] and according to averaged basic anthropomorph logical characteristics [17]. The propulsive efficiency of the body and hydrodynamic endurance, which are various components of passive endurance, determine the overall swimming efficiency. The shape and morphology of the swimmer’s body, body density, and body position in the water are passive components of aerodynamic drag or depend on various aspects of passive and active aerodynamic drag. Different forms of water resistance, such as the sum of the overall resistance, are affected by these factors during swimming as passive components of resistance or depend on a variety of global factors such as frictional resistance, frontal area, and body shape, as well as wave resistance as active components [101]. Thus, the term “BC” refers to the relative proportions of all major body components, such as fat, bone, muscle, and water mass [88].

Compared to previous findings, female swimmers show greater height, lower body weight, and lower BMI [17]. Identifying the ideal balance between body characteristics, such as lean mass and fat mass indices, is likely to be beneficial and of some importance in maximising swimming performance [97]. Thirty years ago, one of the most important studies on the link between the somatotype and BC performance of competitive swimmers was published [67]. The authors found early in the season, only in female swimmers, a statistically significant relationship between performance and body height, body fat percentage, and fat-free weight. Higher correlations were found between performance variables and body height, fat-free weight, body weight, and ectomorphic and mesomorphic body types during the main period of the season. The authors concluded that only female athletes can use BC and somatotype characteristics as predictors of performance. Correlations between PFFM (percentage fat-free mass) or PFM (percentage body fat), among others, have shown the greatest differences between male and female characteristics [157].

About these aspects, it has been observed that muscle size is strongly related to strength and power [73,88]. As a result, FFM allows athletes to develop more muscle strength in specific exercises, which increases speed, quickness, acceleration, and agility [75,82]. The amount of skeletal muscle mass in the body determines the importance of the body structures that contribute to the contractile potential of swimmers, regardless of gender. The indices of muscle and adipose tissues reflect the FFM of the body. These variables show the ability of the swimmer’s bodies to generate force and power. Through the mechanisms of increased stroke traction force and stroke efficiency mechanisms, strength parameters have been suggested as one of the most significant specific factors positively influencing swimming performance [92,97,165]. As a result, a significant improvement in body strength or segmental body strength in swimmers (arms, legs, or trunk) leads to higher force per stroke, as well as higher sprint and turn speeds and better performance [86,87]. So, longitudinal anthropometric characteristics, such as the body height and length of upper (arm span) and lower limbs, appear to be crucial for goal achievement [83,100].

So, as swimming requires energy to maintain the body above water and develop the muscle strength necessary to overcome water resistance, a reduction in the amount of body fat is likely to have an impact on body shape (frontal area) and skin resistance to friction [73]. In addition, there is significant evidence that fat reduction increases muscular and CE, as well as speed and agility [88,97,101].

Several studies (Table 7) have looked for anthropometric, technical, and physiological markers related to swimming performance [74,166,167]. One of these studies [166], revealed seven ‘common’ characteristics that benefited all swimmers, suggesting that swimmers benefit from having less FM, broad shoulders and hips, a longer arm span, and a larger forearm circumference with a smaller relaxed arm circumference.

Anthropometric characteristics are known to play an important role in talent identification and development and have an impact on swimming performance [98,168]. For example, it has been found [95] that LM is the most significant whole-body characteristic, and that having greater limb segment length ratios (i.e., arm (lower arm)/(upper arm) and foot-leg ratio = (foot)/(lower leg)) was essential for achieving higher swim speeds [98]. The importance of arm span, stroke length, and propulsive efficiency in relation to performance in swimmers has also been demonstrated.

The only ‘whole-body’ size characteristic that had the greatest impact was %FM. The findings of the studies conducted [103] broadly support the findings of previous research that has identified anthropometric variables, including FM (%), as important predictors of swimming performance [74,85,93]. This at least partly reflects FM and FFM in terms of the swimmers’ BC [78,80,81]. These variables have an impact on the swimmers’ performance. Large numbers of muscle mass do not appear to have improved swimming performance because they would probably decrease buoyancy and performance [84,91]. However, previous studies [95] support the idea that having a higher FFM increases swimming performance, as it is clearly shown that FFM significantly influenced the prediction of propulsive force, which in turn may result in better swimmer performance. Since the muscles involved must exert more force and thus consume more energy, a longer lever length (arm or leg) is potentially mechanically unfavourable. However, a longer lever length allows greater distance and range to generate propulsion, which reduces the greater energy requirement by using fewer strokes [95,168].

Previous research has also shown that a longer arm span improves swimming efficiency due to greater arm length [91]. The advantage of having a longer wingspan is obvious because this segment functions as a paddle and gives the swimmer more power to move forward through the water. The length of the lever increases the range and distance available to generate propulsion, counteracting the increased power by using fewer strokes [91,95].

This ability to produce propulsive force from a larger area of muscle remains uncertain. However, it has been found [169] that force production capacity is considered an important variable. Physical characteristics such as height, arm span, BC, and somatotype can also affect the level of performance. These morphological traits can significantly affect swimming performance [170] as can be seen in Figure 4.

#### Reference Values for Body Composition in Swimmers

Despite the significant increase in participation in ultra-endurance swimming events, studies specific to open water swimming (OWS) remain scarce. The main difficulty in OWS research is creating standardised study conditions, as it is not possible to reproduce the challenges of such an ultra-endurance race in a laboratory [172]. While there are several studies on the physical and physiological characteristics of OW swimmers, to the authors’ knowledge, there are not much data on the physiological responses and nutritional strategies during ultra-endurance swimming events in swimmers, and their relationship with BC and how it affects performance in terms of propulsion, energy expenditure, and swimming efficiency [173].

Although there is no ideal body fat percentage value for every sport, every athlete may have an ideal body fat percentage value, i.e., a body fat percentage that is related to greater athletic ability and higher scores. The values for the sum of seven folds in top Australian swimmers are shown in Table 8 [105].

However, in terms of BC, studies show a significant and important relationship between BC variables and the performance of both male and female swimmers [157]. For men, optimal swimming performance is more associated with an adequate balance between contractile and non-contractile tissue, and an optimal level of muscle [97]. For women, the key factor is an optimal level of muscle with an appropriate level of fat [7]. In addition, regression analyses show that it is possible to predict swimming performance from BC variables [104].

Other studies have analysed theoretical aspects related to energy balance in OWS swimming competitions, highlighting the specific performance and physiological patterns of this sport modality [173]. Some important characteristics have been identified, mainly in average swim speed and displacement rate, which involve changes in energy metabolism, affecting VO_2_, heart rate, and blood lactate levels [174].

As for propulsive efficiency and hydrodynamic resistance (drag force), these are the main factors that determine the energy cost of swimming (or aquatic locomotion in general) [45]. The problem is that these parameters are difficult to measure in the aquatic environment, but recent studies suggest that estimates of propulsive efficiency and hydrodynamic resistance are quite accurate, allowing the energy cost to be calculated with good precision [175]. In addition, there is a large consensus on methods for estimating metabolic power, which is used to calculate energy cost [159]. On the other hand, several factors, such as gender, age, and training, are known to influence energy cost through their effects on propulsive efficiency and hydrodynamic endurance [45]. But to isolate the effects of a particular factor, it is necessary to keep the other ‘confounding factors’ constant, which is not always easy to do or to calculate [176], which may explain some of the discrepancies observed in the literature on the determinants of energy cost with the goal of increasing energy expenditure. This will mean that the %FM of swimmers does not vary due to errors, variations, ethnic differences, or individual variability [177], and that the %FM and skinfold sum values given in Table 8 are used as an indicator, not a target [178].

### 4.4. Nutrition

The nutritional needs of open water swimmers during competition differ from those of pool swimmers, as the duration of each event, environmental conditions, and nutrient intake determine the requirements. A well-planned strategy, therefore, ensures optimal performance (Table 9).

Swimming, like other aquatic sports, presents significant nutritional challenges in both training and competition. Energy and carbohydrate needs can vary between swimmers and at contrasting times of the week, macrocycle, annual program, or race swim. For the best results, swimmers must learn to adjust their food intake accordingly, especially to provide optimal nutritional support before, during, and after training. Achieving ideal fitness can be particularly challenging for female swimmers. Nutrition during competition requires a special meal plan to promote recovery between races, especially for swimmers participating in multiple events [19].

Swimmers can use various supplements and sports foods to achieve their nutritional goals and optimal performance. Many of the recommendations for swimmers are based on research evidence from other sports with similar characteristics. More studies specific to swimming are needed to support or develop new guidelines [6].

In long-distance swimming, glycogen stores can be depleted. It has already been shown [179] that high-intensity swimming of only 5500 m significantly depletes muscle glycogen in type I and II muscle fibres [179], which reduces stroke distance efficiency [115], so protecting endogenous CHO stores is crucial for performance as stroke efficiency is an important factor in energy expenditure in open water swimming [18]. Thus, CHO loading in combination with the gradual tapering of exercise appears to be the best strategy for further super-compensation of muscle glycogen stores, as this form of work has been found to improve endurance and performance in races lasting longer than 90 min [180]. In conventional CHO loading regimens, a prior CHO depletion phase lasts for an extended period of 7 days before CHO replenishment is initiated [181].

According to more recent research, it is possible to replenish muscle glycogen stores by consuming 10–12 g/kg BM/day of CHO in the 36–48 h before exercise, in conjunction with regular training [182]. To ensure adequate ‘loading’, swimmers are advised to increase CHO intake and energy intake [183]. Maintaining a high average pace is useful to combat fatigue with these strategies, especially in the final phases of endurance races of more than two hours [180]. The benefits of CHO may become less evident as the duration of a race increases (25 km or more), or at least should be supplemented with exogenous sources of CHO.

In terms of ensuring adequate hydration, the prerace meal should normally consist of easily tolerated foods with a CHO target of 1–4 g/kg BM 1–4 h before the start of the race [184], along with an adequate number of fluids, and always remembering that the swimmer should continue to consume fluids or snacks containing CHO until the race begins.

In general, swimmers in long-distance events tend to follow similar nutritional strategies to endurance athletes such as runners, cyclists, or triathletes. However, it should be noted that long-distance swimmers may have specific nutritional needs due to the unique characteristics of their sport. For example, the hydrostatic pressure of the water and the resistance offered by the aquatic environment may affect nutrient absorption and metabolism differently from land-based sports [173,185].

Optimal prerace nutritional strategies should, therefore, seek to increase CHO, fat, electrolyte, and fluid intake over the course of a race, as it is crucial to maintain an adequate balance of these nutrients, because as the duration of long-distance races increases, athletes increase fluid intake, which increases the risk of exercise-associated hyponatraemia and limb swelling [186,187].

To reduce these risks, fluid intake should be limited to 300–600 mL per hour and supplemented with electrolytes such as sodium and potassium. In addition, adequate carbohydrate and fat intake before and during competition will help maintain energy levels and prevent exhaustion. It is important for long-distance swimmers to work closely with sports nutrition experts to develop a nutrition and hydration plan tailored to their needs [188,189].

Elite swimmers expend a lot of energy due to their extensive training regimen, and they must ensure they consume enough food to meet this high energy demand. Alongside having a proper hydration plan before, during, and after training to enhance performance and recovery, taking supplements in the appropriate amounts can also boost an athlete’s health, athletic abilities, and recovery process [3].

### 4.5. Ergogenic Nutritional Aids

The consumption of vitamins and minerals during training and competition is a common nutritional practice in this type of event [190]. Therefore, it is not surprising that the effect of vitamins, minerals and other ergogenic supplements on ultra-endurance athletes has been studied [191]. It should be noted that vitamin C and E intake could potentially decrease training adaptations [192].

Among the micronutrients, iron deficiency is most likely, and inadequate iron status can impair exercise performance through suboptimal haemoglobin levels and changes in the muscle, including a reduction in myoglobin and iron-related enzymes [193].

Exercise has a major influence, altering plasma volume and function response to exercise stress and fatigue, resulting in reduced blood haemoglobin concentrations due to the expansion of plasma volume in response to endurance and ultra-endurance exercise. This deficiency, known as sports anaemia, does not impair performance [194]. It is difficult to conclude whether this deficit is a true deficiency in iron status that impairs health and performance or sports anaemia. Monitoring the nutritional habits of athletes and providing them with adequate doses of bioavailable iron in food would help prevent true deficiency. Ferritin levels < 20 ngmL^−1^ are a reliable indicator for diagnosing true anaemia. Athletes must take supplements under medical guidance to replenish the amount lost and avoid consuming excessive amounts of iron, as this can lead to haemochromatosis.

In terms of the use of performance-enhancing aids in swimming, there has been [24] particular interest for open water swimmers in products that can improve central drive and help buffer acidic environments and translate this into increased performance, with substances such as those listed below.

#### 4.5.1. Sodium Bicarbonate

Numerous studies have delved into the effects of sodium bicarbonate (NaHCO_3_) on performance in various forms of exercise, including endurance (Table 10). The effects of different sodium bicarbonate ingestion protocols to optimise ergogenic effects while simultaneously reducing the occurrence and severity of side effects have also been examined [195].

This cannot be confirmed with certainty, as the results from the literature are very inconsistent and difficult to compare. For this reason, the literature does not provide solid evidence on the exact effects of NaHCO_3_ (Table 11). In addition, the side effects of NaHCO_3_ may have hurt performance, although long-term administration protocols may overcome these circumstances.

Based on the existing literature, we can qualify the use of NaHCO_3_:Taking sodium bicarbonate (in doses of 0.2 to 0.5 g/kg) improves performance in muscular endurance activities such as swimming [219].NaHCO_3_ improves performance in single and multi-session exercise [219].NaHCO_3_ improves exercise performance in both men and women [220].For single-dose supplementation protocols, 0.2 g/kg sodium bicarbonate appears to be the minimum dose necessary to experience improvements in exercise performance. The optimal dose of sodium bicarbonate for ergogenic effects appears to be 0.3 g/kg. Higher doses (e.g., 0.4 or 0.5 g/kg) may not be necessary as they do not provide additional benefits and are associated with a higher incidence and severity of adverse side effects [221].For single-dose supplementation protocols, the recommended time to take sodium bicarbonate is 60–180 min before exercise or competition [219].Multiple days of sodium bicarbonate supplementation can improve exercise performance. These protocols typically last 3–7 days prior to activity, and a total daily dose of 0.4 or 0.5 g/kg of sodium bicarbonate produces beneficial effects. The daily dose is divided into several intakes throughout the day, such as breakfast, lunch, and dinner. The advantage of multi-day protocols is that they can help reduce the potential side effects of sodium bicarbonate on the day of competition [222].Prolonged use of sodium bicarbonate (such as before each exercise session) may improve training adaptations, such as increasing time to fatigue and power output [219].The most common side effects of sodium bicarbonate supplementation are bloating, nausea, vomiting, and abdominal pain. The frequency and severity of these effects vary from person to person and from individual to individual but are generally mild. Nevertheless, these side effects can adversely affect exercise performance. Strategies to minimise the likelihood and severity of these effects include taking smaller doses (such as 0.2 g/kg or 0.3 g/kg), taking approximately 180 min before exercise or adjusting the timing according to individual response, taking with a carbohydrate-rich meal, and using enteric-coated capsules [223].Combining sodium bicarbonate with creatine or beta-alanine may have additional effects on exercise performance. It is unclear whether combining sodium bicarbonate with caffeine or nitrates provides additional benefits [219].

Research has also explored the interaction of sodium bicarbonate with other ergogenic aids such as beta-alanine, caffeine, and creatine.

#### 4.5.2. Sodium Bicarbonate and Beta-Alanine

The simultaneous use of sodium bicarbonate and beta-alanine is one of the best-researched supplement combinations (Table 12). Beta-alanine is an effective performance enhancer that must be taken over a long period (e.g., several weeks) to increase muscle carnosine content, thereby improving the muscle’s buffering capacity [224]. Since beta-alanine increases intracellular pH buffering and sodium bicarbonate increases extracellular pH buffering, it seems reasonable to consider that the combination of the two would have an additive ergogenic effect. Other studies have also suggested potentially greater improvements by combining sodium bicarbonate and beta-alanine compared with sodium bicarbonate alone in swimming [225]. A meta-analysis demonstrated that adding sodium bicarbonate to beta-alanine supplementation produced greater ergogenic effects than beta-alanine alone (pooled Cohen’s d: 0.43 vs. 0.18) [224]. Overall, the findings suggest that concurrent beta-alanine and sodium bicarbonate supplementation can generate potentially significant increases in exercise performance over-supplementation with either alone.

Preliminary studies indicate that beta-alanine can effectively decrease lipid oxidation and attenuate the accumulation of harmful compounds when combined with aerobic physical activity in both men and women [227,228].

According to the findings [229], beta-alanine supplementation significantly improved ventilatory threshold and time to exhaustion but had no effect on VO_2max_. Although no statistically significant difference was found, the data suggest that beta-alanine most likely has a beneficial effect on time to exhaustion [230]. In addition, both peak oxygen consumption and power output at the ventilatory threshold increased similarly with and without beta-alanine supplementation, which has not been observed in other studies [231].

The recommended dose is 1–3 g per day, although in some studies, more than 6 g per day were administered [232]. No information on toxic or carcinogenic effects has been found [233]. The only side effects reported are short-lived tingling sensations and a slight increase in alanine aminotransferase levels [234]. Supplementation with 6.4 g per day for 24 weeks did not significantly affect clinical markers of kidney, liver, and muscle function, nor did it produce chronic sensory side effects [232].

#### 4.5.3. Sodium Bicarbonate and Caffeine

Caffeine is a well-established ergogenic aid, with recorded performance benefits for aerobic and muscular endurance, power, jump height, and muscular strength [235,236,237]. The ergogenic effects of caffeine are generally explained by its ability to act as an adenosine receptor antagonist, which helps to reduce fatigue, pain, or perceived exertion [236]. Since sodium bicarbonate and caffeine may enhance performance through different mechanisms, their simultaneous use could produce additive effects. However, it should be noted that the supplementation protocols in some of the studies [198] resulted in a high incidence and severity of side effects and were also ineffective.

#### 4.5.4. Sodium Bicarbonate and Creatine

Creatine is a well-studied ergogenic supplement [238]. Numerous studies have investigated the combined effects of sodium bicarbonate and creatine and have reported an additive effect.

#### 4.5.5. Sodium Bicarbonate and Nitrates

The combination of sodium bicarbonate and nitrates may be counterproductive due to their mechanisms of action. Alkalosis caused by bicarbonate supplementation may reduce the efficacy of nitrate supplementation. This is because the conversion of nitrate to nitric oxide is facilitated by an acidic environment [239].

## 5. Conclusions

In long-distance swimming, meeting energy needs is essential to achieve competitive goals. In swimmers, muscular strength enables them to produce more intense propulsion during swimming movements, which gives them a superiority over their competitors. In addition, increased muscle mass in the trunk and limbs enhances stability and body control in the water, crucial elements for an effective swimming method. Furthermore, anaerobic strength, i.e., the ability to generate large volumes of energy in short periods, is vital in activities where high acceleration and maximum speed are required. Thus, the integral development of general and specific strength in swimming enables athletes to improve their performance in competition. These general and specific strength skills play a crucial role in the final performance of swimmers. Swimmers with an ideal constitution and an adequate body structure usually possess an adequately developed muscle mass in the trunk and limbs. This strong musculature is linked to anaerobic strength, both in the dry and aquatic environment. Given the emerging nature of these tests, there is little research available to inform nutrition practices. Current nutritional suggestions for open water training and competition extend beyond pool suggestions or extend from other athletic communities with similar physiological needs. In competition nutrition should focus on improving pre-competition hydration and glycogen stores. During competition, swimmers should rely on their own sources of fuel and fluids, making use of feeding areas only when tactically appropriate, in addition to the proper periodisation of supplements.

The main findings are that there is convincing evidence that various physical, technical, and functional factors are linked to swimming performance and body composition, as each athlete may have an ideal level of body fat that relates to improved performance and results. Thus, the nutritional needs of open water swimmers during competition are vastly different from those of pool swimmers.

## Figures and Tables

**Figure 1 nutrients-16-03949-f001:**
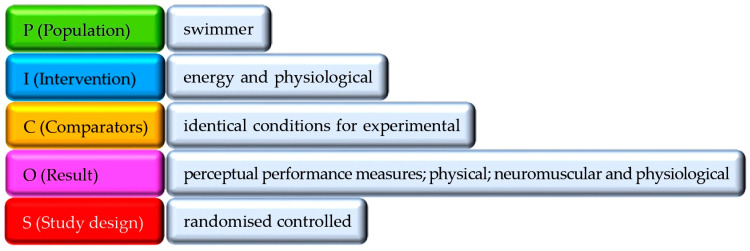
PICOS models.

**Figure 2 nutrients-16-03949-f002:**
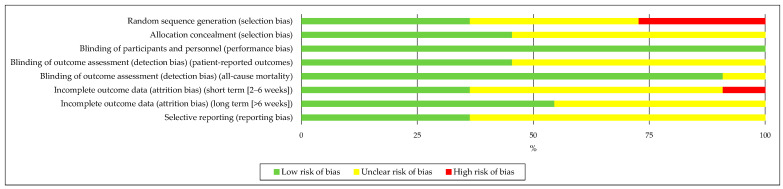
Risk of bias graph. Green for negligible risk of bias, yellow for unclear risk of bias, and red for substantial risk of bias. Shows the overall risk of bias for each domain. For example, the length of the green rectangle means the number of studies assessed as minimal risk of bias.

**Figure 3 nutrients-16-03949-f003:**
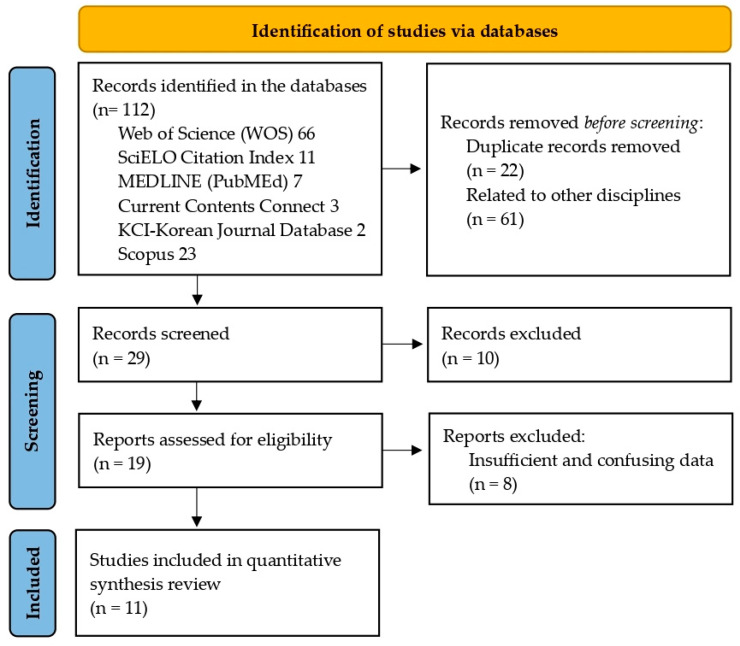
Study selection flowchart [49]. (This work is licensed under CC BY 4.0. To view a copy of this license, visit https://creativecommons.org/licenses/by/4.0/ (accessed on 30 October 2024)).

**Figure 4 nutrients-16-03949-f004:**
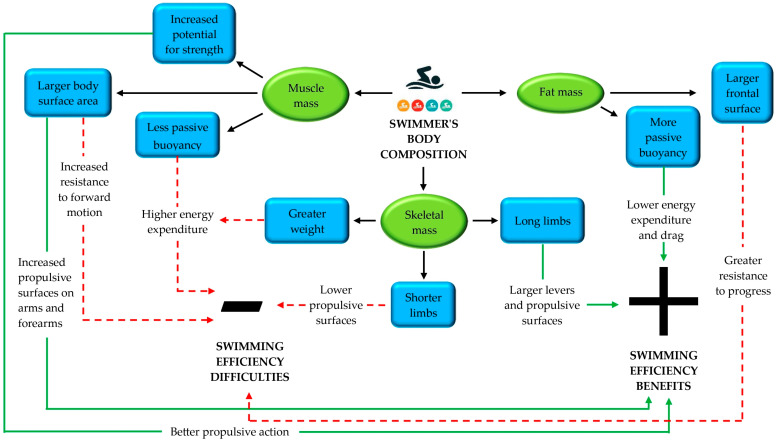
Influence of the swimmer’s BC on swimming efficiency, according to Navarro et al. [171].

**Table 1 nutrients-16-03949-t001:** Summary of risk of bias, indicating the risk of bias for each domain in each study.

	Random sequence generation (selection bias)	Allocation concealment (selection bias)	Blinding of participants and personnel (performance bias)	Blinding of outcome assessment (detection bias) (patient-reported outcomes)	Blinding of outcome assessment (detection bias) (all-cause mortality)	Incomplete outcome data (attrition bias) (short-term [2–6 weeks])	Incomplete outcome data (attrition bias) (long term [>6 weeks])	Selective reporting (reporting bias)
Zamparo 2005 [18]	+	+	+	+	?	+	?	+
Knechtle 2008 [41]	?	?	+	+	+	+	?	+
Knechtle 2010 A [42]	?	?	+	+	+	+	?	?
Knechtle 2010 B [43]	?	?	+	+	+	+	?	+
Pyne 2014 [2]	+	+	+	?	+	?	+	+
Shaw 2014 [19]	+	+	+	?	+	?	+	?
VanHeest 2014 [44]	+	+	+	?	+	?	+	?
Domínguez 2017 [7]	-	?	+	?	+	?	+	?
Zamparo 2020 [45]	-	?	+	?	+	?	+	?
Jiménez-Alfageme 2022 [25]	?	+	+	+	+	?	+	?
Ben-Zaken 2022 [1]	-	?	+	?	+	-	?	?

Green represents a negligible risk of bias; yellow represents an unclear risk of bias; red represents a substantial risk of bias.

**Table 2 nutrients-16-03949-t002:** Levels of evidence of selected studies according to the Oxford quality scoring system [46].

Research	Level
Zamparo et al. (2005) [18]	A/1b
Knechtle et al. (2008) [41]	A/1b
Knechtle et al. A (2010) [42]	A/1b
Knechtle et al. B (2010) [43]	A/1b
Pyne et al. (2014) [2]	A/1a
Shaw et al. (2014) [19]	A/1a
VanHeest et al. (2014) [45]	A/1b
Domínguez et al. (2017) [7]	A/1a
Zamparo et al. (2020) [46]	A/1a
Jiménez-Alfageme et al. (2022) [25]	A/1b
Ben-Zaken et al. (2022) [1]	A/1b

**Table 3 nutrients-16-03949-t003:** Studies included in the work.

	Journal	Q	Authors (Year) and Reference	Population	Method	Intervention	Variables	Outcomes Analysed	Main Conclusions
Ph	Eur. J. Appl. Physiol	Q1	Zamparo et al. (2005) [18]	5 ♂ and ♀ 5elite	S; DB	3 × 400 m + 2 km	VO_2_; BLC; HR; BW; s; SR; WT	EC ± ♂ lower ♀	EC  with s; VO_2_  sEC ♀ < ♂; ST *p* on Eff/P
Anthr	Anthropologischer Anzeiger	Q2	Knechtle et al. (2008) [41]	12 ♂	S; DB	12 h	BMI; BH; LC; SMM; FFM	Average distance covered	AntrC no *p* on P
Anthr	Percept. Mot. Skills	Q3	Knechtle et al. (2010) [42]	39 ♂24 ♀	S; DB	26.4 km	BM; BH; SF; lL; %FM; HCV (km/t); I (km/h)	BH/BMI/Al r with t ♂No r with P and AnthrC	AntrC nor r in tts r with t on ♀/♂; BMI on ♂ too
Anthr	Hum. Mov.	Q2	Knechtle et al. (2010) [43]	15 ♂	S; DB	26.4 km	AntrC; BM; BH; BMI; lL; %FM; Σ7SF; YAS; tV (t/km); ts; TRt; Rs	TRt; Rsr: AnthrC+tv/TRt	ts predictor P ♂AnthrC no *p* with P
Ph	Int. J. Sport Nutr. Exerc. Metab.	Q2	Pyne et al. (2014) [2]	♂ and ♀elite	DB	Seasonal	AnthrC; Ee; Rl; I; Gl	BH; FFM; Pw; aPw; E; VO_2_; BLC	Rs determined by medium, style employed and methods used
Ph	Int. J. Sport Nutr. Exerc. Metab.	Q2	Shaw et al. (2014) [19]	♂ and ♀elite	S; DB	Seasonal	BC; FFM; Gl; PhC; Erc; ErA; VO_2max_	FFM;  VO_2max_; OpGl; Cff	sNrfRp
Nt	Med. Sci. Sports Exerc.	Q1	VanHeest et al. (2014) [44]	10 ♀Elite	DB	12 weeks	Bv; AnthrC; s; EI; EA	BH; BW; BC; MS; EI; EA; H; P	Dp is crucialH r with P
Nt	J. Exerc. Nutr. Biochem.	Q3	Domínguez et al. (2017) [7]	♂ and ♀	S; DB	Seasonal	CHOI; PI; FI; Hc; ErA	CHO; Pt; F; Hr; Cff; Cr; B; β-a; N; vD; Bc; HMB	P  with Hs and Erg
B	Eur. J. Appl. Physiol.	Q1	Zamparo et al. (2020) [45]	♂ and ♀	S; DB	Seasonal	Eff sw; Hd; Er; Ep; s and Em; dPbGSl; Ept; BfaaPw	cEff and Ecsswpt; iHr; EC; dPbGSl; tv; BfaaPw	Eff y Ec  by iPsaEC varies AS; dPbGSltv  Eff and PImportance mBMch
Nt	Nutrients	Q1	Jiménez-Alfageme et al. (2022) [25]	103 ♂ and 29 ♀	S; DB	Seasonal	Sf; Ms; ErA	Sf; Ms; ErA	Sf; Ms; ErA to  P
G	Biol. Sport	Q1	Ben-Zaken et al. (2022) [1]	♂ and ♀	DB	-	Gp	Ra-s; A-ce; AIIp; K-ks; Br; Amd; Rα; Vegf; hACTN3g; Ltg; M; SNPIL6–174G/C	Importance Gp on environmental and physiological demands to  P

♂: male; ♀: female; A-ce: angiotensin-converting enzyme; AIIp: angiotensin II production; Al: arm length; Amd: adenosine monophosphate deaminase; AnthrC: anthropometric characteristics; aPw: aerobic power; AS: athlete’s skill; B: bicarbonate; BC: body composition; Bc: bovine colostrum; BfaaPw: biomechanical factors affecting athletic performance in the water; BH: body height; BLC: blood lactate concentration; blue: target; BM: body mass; BMI: body mass index; Br: bradykinin; Bv: blood variables; BW: body weight; cEff and Ecswpt: comparisons of efficiency and economy in swimming and other aquatic; Cff: caffeine; CHO: carbohydrate; CHOI: CHO intake; Cr: creatine; Dp: dietary periodisation; dPbGSl: differences in performance between genders and skill levels; E: endurance; EA: energy availability; EC: energy cost; Eff: efficiency; EI: energy intake; Em: economy of movement; Ep: efficient propulsion; Ept: effects of propulsion tools; Er: energy requirements; ErA: ergogenic aids; Erc: energy recommendations; F: fats; FFM: fat-free mass; FI: fat intake; FM: fat mass; fRp: flexibility in career race plans; Gl: glucose levels; Gp: genetic polymorphism; green: improvement; H: hormones dietary periodisation; hACTN3g: Human ACTN3 gene; HC: hip circumference; Hc: hydration control, Hd: hydrodynamic drag; Hs: hydration strategy; HMB: β-hydroxy-βmethylbutyrate; HR: heart rate; Hr: hydration requirements, I: intensity, iHr: impact of hydrodynamic resistance; iPsa: increase in propulsive surface area; K-ks: kallikrein–kinin system; LC: limb circumference, lL: length of limbs; Ltg: lactate transport genes; M: Myostatin, mBMch: movement biomechanics; Ms: medical supplements; MS: muscle strength; N: nitrate; Nt: nutrition; P: performance; *p*: significant; PhC: physiological characteristics; PI: protein intake; Pt: protein; Pw: power; Rs: results; Ra-s: renin–angiotensin system; Rl: race length; Rs: race speed; Rα: receptor alpha; s: speed; S: strength; SF: skin folds; Sf: sport foods; SMM: skeletal muscle mass; sNr: specific nutritional recommendations; SR: stroke rate; ST: swimming technique; sw: swimming; t: time; TRt: total race time; ts: training speed; tv: training variables; tV: training volume; UL: upper limbs; V: volume; vD: vitamin D; Vegf: vascular endothelial growth factor; WT: water temperature; YAS: years active swimmer; yellow: possibility; β-a: β-alanine.

**Table 4 nutrients-16-03949-t004:** Energy requirements of endurance-trained swimmers.

Studies	Energy Requirement
Medbø and Tabata, 1989 [50]	40%
Withers et al., 1991 [51]	28%
Ring et al., 1996 [52]	29%
Bogdanis et al., 1996 [53]	34%

**Table 5 nutrients-16-03949-t005:** Summary of energy balance in swimmers from self-reported energy intake and estimated energy expenditures.

Study	Methods for Measuring EI or EE	*n*	EI MJ/day (kcal/day)	EE MJ/day (kcal/day)
Vallieres et al., 1989 [54]	EI: 3 day food diary Weeks 1 and 4 EE: Activity records, RMR and swimming EE via indirect calorimetry	6 ♀	Week one 10.4 ± 3.2 (2500 ± 800) Week four 10.2 ± 2.9 (2400 ± 700)	Week one 12 ± 2.4 (2900 ± 600) Week four 10.5 ± 1.5 (2500 ± 400)
Jones and Leitch 1993 [55]	EE: Doubly labelled water EI: Standardised diet provided and correlated, to detailed weighed food records	5 ♂ 3 ♀	16.3 ± 2.6 (3895 ± 621)	14.6 ± 3.7 (3556 ± 1025)
Almeras et al., 1997 [56]	EE: Physical activity record (3 days) and HR-VO_2_ method (2 day, 24 h HR measurement) EI: 3 day food diary	11 ♀	Baseline 11.0 ± 3.3 (2600 ± 800) Part 1 10.5 ± 3.4 (2500 ± 800) Part 2 11.1 ± 3.0 (2600 ± 700) Part 3 9.2 ± 3.0 (2200 ± 700)	PAR 12.0 ± 3.1 (2900 ± 700) PAR 12.0 ± 2.7 (2900 ± 600) HR-VO_2_ 12.7 ± 6.4 (3000 ± 1500) PAR 13.2 ± 2.4 (3100 ± 600) HR-VO_2_ 12.4 ± 3.6 (2900 ± 900) PAR 12.7 ± 2.2 (3000 ± 500) HR-VO_2_ 13.1 ± 4.0 (3100 ± 1000)
Trappe et al. 1997 [57]	EE: Doubly labelled water EI: 2 day food diary	5 ♀	13.1 ± 0.9 (3136 ± 227)	23.4 ± 2.1 (5593 ± 495)
Ousley-Pahnke et al., 2001 [58]	EE: Activity diary and equations EI: 4 day food diary	15 ♀	9.5 ± 2.8 (2275 ± 665)	9.8 ± 0.7 (2342 ± 158)
Hassapidou and Manstrantoni 2001 [59]	EE: 7 day activity records and equations EI: 7 day weighed dietary records	9 ♀	Training: 8.4 ± 2.3 (2015 ± 542) Competition: 7.9 ± 3 (1890 ± 709)	Training: 10.5 ± 1.3 (2520 ± 304) Competition: 10.7 ± 0.9 (2550 ± 210)
Sato et al., 2011 [60]	EE: RMR, activity factor, and VO_2_ estimations EI: 3 day food records	6 ♂ 13 ♀	Preparation: ♂: 13.1 ± 3.1 (3158 ± 733) ♀: 11.3 ± 1.8 (2710 ± 431) Intense ♂: 13.9 ± 1.6 (3322 ± 378) ♀: 12.0 ± 1.7 (2880 ± 408)	Preparation: ♂: 11.1 ± 0.4 (2646 ± 146) ♀: 8.7 ± 1.4 (2085 ± 326) Intense ♂: 12.3 ± 1.4 (2932 ± 335) ♀: 11.3 ± 1.8 (2562 ± 372)
Slattery et al. 2012 [61]	EI: 4 × 24 h recall EE: HR calculation	4 ♂ 2 ♀	16.6 ± 3.4 (3995 ± 789)	16.7 ± 3.3 (3969 ± 821)

EI: energy intake; EE: energy expenditure; HR = heart rate; VO_2_ = oxygen consumption; RMR = resting metabolic rate; PAR = physical activity record.

**Table 6 nutrients-16-03949-t006:** Protein intake.

Research	Intake
Burke, 2007 [62]	+ than 1.2–1.6 g/kg body weight/day
Moore et al., 2009 [63]	~20–25 g high quality protein
Witard et al., 2014 [64]	

**Table 7 nutrients-16-03949-t007:** Studies related to swimming performance.

Research	Observed Performance Parameters
Stager et al., 1984 [65]	BH, BM, FFM, and rLV showed statistically significant correlations
Chatard, et al., 1990 [66]	A larger body and a larger surface area will increase drag, associated with a lower running s for a given amount of MP
Siders et al., 1993 [67]	Statistically significant relationship between P and BH, % FM, and FFM only in ♀ swimmers. They found higher correlations for the same variables of P and BH, FFM, BW, and ectomorphic and mesomorphic BT
Lowensteyn et al., 1994 [68]	Body resistance characteristics could explain the effect of FM variables on ♀ performance
Pelayo et al., 1996 [69]	Longitudinal AnthrC, such as BH and length of the UL (arm span) and LL are of paramount importance to achieve high results
Strass et al., 1998 [70]	Increased FFM allows more MS to be produced during the sp movement
	The Bs is influenced by the SMM, the FFM, and the strct relationships between the MTI and ATI
	It has been suggested that S parameters are one of the most crucial sp factors that positively influence sw P by increasing the pull force of the stk and improving stk efficiency
Dopsaj et al., 1999 [71]	Significant improvement in BS or sBS (arms/legs/trunk) results in greater max force per stk
Nevill, 2000 [72]	Swimmers benefit from having less FM, a longer (but shorter) arm span and a larger forearm circumference with a smaller relaxed arm circumference
Mameletzi et al., 2003 [73]	Lower FM probably translates into lower aerodynamic drag (frontal area) and frictional drag. Body contractile BC provides better propulsive force potential
Geladas et al., 2005 [74]	Only % FM was the single most important “whole body” size characteristic
Barbosa et al., 2006 [75]	An increase in FFM allows more MS to be produced during sp movement efforts
Pyne et al., 2006 [76]	MM appears to correlate with a high level of S and propulsion
	FFM and TBW as absolute indicators of BC as predictors of best P
Brauer et al., 2007 [77]	Current swimmers tend to be taller than in the past
Jürimäe et al., 2007 [78]	FM and FFM appear to contribute to swimmers’ performance
Barbosa et al., 2010 [79]	The Bs is influenced by the SMM, the FFM, and the strct relationships between the MTI and ATI
	It has been suggested that S parameters are one of the most crucial sp factors that positively influence sw P by increasing the pull force of the stk and improving stk efficiency
Saavedra et al., 2010 [80]	FM and FFM appear to contribute to swimmers’ P
Lätt et al., 2010 [81]	FM and FFM appear to contribute to the P of swimmers
Saavedra et al., 2010 [80]	
West et al., 2011 [82]	Increased FFM allows more MS to be produced during sp movement efforts
Kjendle et al., 2011 [83]	AnthrC, such as BH and length of the UPL (arm span) and LL are of primary importance to achieve high results
Pérez et al., 2011 [84]	P was apparently not helped by the large MM values
West, et al., 2011 [82]	Increasing muscle or FFM allows more MS to be produced during sp movement efforts
Zuniga et al., 2011 [85]	AnthrC, including FM (%), as important predictors of P
	The %BF was the most important size characteristic
Morouço et al., 2011 [86]	Significant improvement in sBS (arms, legs, or trunk) results in greater max force per stk
Morouço et al., 2011 [86] Morouço et al., 2012 [87]	The Bs is influenced by the SMM, the FFM, and the strct relationships between the MTI and ATI
	It has been suggested that S parameters are one of the most significant sp factors that positively influence sw P by increasing the pulling force of the stk and improving stk efficiency
Morouço et al., 2012 [87]	Significant improvement in segmental BS (arms/legs/trunk) results in greater max force per stk
Ratamess et al., 2012 [88]	Reducing FM contributes to muscular and CE, as well as to the development of s and agility
Santos et al., 2014 [89]	International swimmers, both ♂ and ♀, are taller, with less FM, lower BMI, but with a higher level of MM than national level swimmers
Copic et al., 2014 [90]	Increased FFM produces more MS, which improves s, quickness, acceleration, and agility
Moura et al., 2014 [91]	sw P was apparently not helped by large MM values because they were likely to reduce buoyancy and impair P
Gatta et al., 2015 [92]	The effect of FM variables on swimmers’ performance could be explained by body resistance
Bond et al., 2015 [93]	AnthrC, including FM (%), as important predictors of sw P, although only FM% was the most important
Nasirzade et al., 2015 [94]	Significant relationship between muscle architectural characteristics; muscle thickness, and triceps brachii fascicle length
Nevill et al., 2015 [95]	FFM was the single most important characteristic associated with sw s
	sw P is associated with changes in size, proportions and BC, as well as biological maturation
	Increased MM improves s P
Gatta et al., 2016 [96]	The Bs is influenced by the SMM, the FFM, and the strct relationships between the MTI and ATI
	It has been proposed that through mechanisms of increased stk pull and stk efficiency mechanisms, sw P is positively influenced
Roelofs et al., 2017 [97]	Identifying the optimal balance between body characteristics: FFM and FM parameters are likely to be beneficial and of some importance in maximising sw P
	The Bs is influenced by the SMM, the FFM, and the strct relationships between the MTI and ATI
	It has been suggested that S parameters are one of the key factors that positively influence sw P by increasing the pull force of the stk and improving stk efficiency
	Reducing FM contributes to muscular and CE, as well as to the development of s and agility
	FFM appears to be logically correlated with a high level of S and propulsion
Morais et al., 2017 [98]	AnthrC play a crucial role in talent identification and development, as well as in sw P
Sammoud et al., 2019 [99]	AnthrC are important factors in identifying developing talent, as well as an influence on sw P
Morales et al., 2019 [100]	AnthrC, such as BH and length of the UL (arm span) and LL are of primary importance to achieve high results
	The importance of AnthrC in sw, the increase in swimmers’ s, the result of the increase in stk length and stk rate
Cortesi et al., 2020 [101]	The max sw s will be reached by the swimmer who can achieve the highest max metabolic power with the lowest energy consumption during the swim
Dopsaj et al., 2020 [102]	Today’s elite, both ♂ and ♀ swimmers are taller, heavier, and bigger than in the past
Cortesi et al., 2020 [101]	The reduction of FM contributes to muscular and CE
	The effect of FM on performance in female swimmers could be explained by body resistance
Dos Santos et al., 2021 [103]	Height and BM did not contribute significantly to the study of variation in anatomical and physiological dimensions in swimmers
Espada et al., 2023 [104]	The body segments (LL, UL, and trunk) and the corresponding tissue content (TM, FM, and FFM+BMC) reinforce the importance of BC assessment in sw

AnthrC: anthropometric characteristics; ATI: adipose tissue indices; BC: body composition; BH: body height; BM: body mass; BMC: bone mineral content; BMI: body mass index; BS: body strength; Bs: body structure; BT: body type; BW: body weight; CE: cardiorespiratory endurance; FFM: fat-free mass; FM: fat mass; LL: lower limbs; MM: muscle mass; MP: mechanical power; MS: muscle strength; MTI: muscle tissue indices; P: performance; rLV: residual lung volume; s: speed; S: strength; sBS: segmental body strength; SMM: skeletal muscle mass; sp: specific; stk: stroke; strct: structural; sw: swimming; TM: total mass; UL: upper limbs.

**Table 8 nutrients-16-03949-t008:** Sum of seven folds (S7P) and body fat percentage values in elite swimmers [105].

S7P ♂ mm	S7P ♀ mm	♂ %FM	♀ %FM
51.9	80.4	8.1	20.5

**Table 9 nutrients-16-03949-t009:** Open water swimming events place a significant emphasis on nutrition.

Event	Length	Physiological Difficulties	Dietary Emphasis	Feeding Approach
5 km	~0 m–1 h	TH	▪ Maximise glycogen storage ▪ Improve hydration before racing in hot weather ▪ Prevent overheating in hot conditions ▪ Use caffeine ▪ Take supplements to boost buffering ability	Minimal
10 km	~1 h 40 m –2 h 10 m	TH GD FO	▪ Achieve ideal glycogen storage ▪ Improve hydration before a race in hot weather; consider precooling methods for hot conditions ▪ Enhance the rate of carbohydrate oxidation ▪ Utilise caffeine ▪ Take supplements to boost buffering capacity.	Floating platforms and on body
25 km	~4–5 h	TH GD FO GI	▪ Achieve ideal glycogen storage ▪ Enhance hydration before a race in hot weather; use precooling techniques in hot conditions ▪ Boost the rate of carbohydrate oxidation ▪ Hot conditions may benefit from consuming warm food and drinks ▪ Utilise caffeine ▪ Consider sodium requirements in hot environments	Floating platforms mainly
>25 km	>5 h	TH GD FO GI	▪ Achieve optimal storage of glycogen ▪ Enhance hydration status before a race in hot weather; employ techniques to cool the body in hot weather ▪ Increase the rate at which carbohydrates are converted into energy ▪ Consuming warm food and beverages may be advantageous in cold weather ▪ Utilise caffeine ▪ Adopt a balanced food and beverage composition to ensure gastrointestinal comfort ▪ Consider sodium requirements in hot environments	Floating platforms, assisting vessels (accessible at intervals of 2.5 km)

**Table 10 nutrients-16-03949-t010:** Sodium bicarbonate (NaHCO_3_) supplementation in endurance.

Research	Athletes	Supplementation (Type/Dose)	Side Effects		Effects of NaHCO_3_
McClung et al., 2007 [196]	Endurance athletes	Fluid solution; 0.3 g/kg BM	Mild	YES	NaHCO_3_ resulted in performance improvement and lowered blood lactate (F_(1,15)_ = 51.4; *p* < 0.001; η^2^ = 0.774)
Lindh et al., 2008 [197]	Elite swimmers	Capsule; 0.3 g/kg BM	None	YES	NaHCO_3_ improved performance in eight out of nine athletes by 1.6% (*p* = 0.04)
Pruscino et al., 2008 [198]	High elite swimmers	Capsule; 0.3 g/kg BM	N/A	NO	No significant improvement in time after ingestion of NaHCO_3_ (ES = 0.25 ± 0.26; *p* = 0.052)
Zajac et al., 2009 [199]	Well-trained swimmers	Fluid solution; 0.3 g/kg BM	None	YES	NaHCO_3_ ingestion improves performance by 1.5 s compared to controls (F_(2,28)_ = 5.63; *p* < 0.05)
Siegler et al., 2010 [200]	University swimmers	Fluid solution; 0.3 g/kg BM	None	YES	NaHCO_3_ improved total swim time by 2%. Mean difference overall was 4.4 s (d = 0.15; *p* = 0.04)
Kilding et al., 2012 [201]	Well-trained cyclists	Capsule; 0.3 g/kg BM	Mild	YES	Caffeine and NaHCO_3_ consumed separately led to performance enhancements (ES = 0.21; *p* = 0.01)
Joyce et al., 2012 * [202]	Competitive swimmers	Capsule; 0.1 g/kg BM	Mild	NO	Chronic supplementation of NaHCO_3_ had no effect on swimming performance (F = 0.48; *p* = 0.8)
Joyce et al., 2012 * [202]	Competitive swimmers	Capsule; 0.3 g/kg BM	Mild	NO	Acute supplementation of NaHCO_3_ had no effect on swimming performance (F = 0.48; *p* = 0.08).
Kupcis et al., 2012 [203]	Lightweight rowers	Capsule; 0.3 g/kg BM	None	NO	NaHCO_3_ provides no benefit for rowing performance (*p* = 0.41; ES: 0.05)
Tobias et al., 2013 [204]	Competitive swimmers	Capsule; 0.3 g/kg BM	Mild	YES/NO	Combined with BA, NaHCO_3_ improved 200 m time (F = 1.36; *p* = 0.28), but not 100 m (F = 5.17; *p* = 0.024).
Mero et al., 2013 [205]	Competitive swimmers	Capsule; 0.3 g/kg BM	None	YES	NaHCO_3_ improves swimming performance by 2.4%/1.5 s (*p* < 0.05)
Mueller et al., 2013 [206]	Cyclists/triathletes	Tablet; 0.3 g/kg BM	N/A	YES	NaHCO_3_ improved time to exhaustion compared to a placebo (+23.5%) (F_(1,7)_ = 35.45; *p* = 0.001 η^2^ = 0.84).
Driller et al., 2013 [207]	National team rowers	Capsule; 0.3 g/kg BM	Mild	NO	Serial supplementation of NaHCO_3_ before HIT provides no benefits to performance (*p* > 0.05)
Hobson et al., 2013 [208]	Well-trained rowers	Capsule; 0.3 g/kg BM	None to Severe	NO	Neither NaHCO_3_ or Beta alanine (or combined) have an effect on performance (*p* < 0.05)
Hobson et al., 2014 [209]	Well-trained rowers	Capsule; 0.3 g/kg BM	None to Severe	NO	Ingestion of NaHCO_3_ has no effect on rowing performance (*p* < 0.09)
Stöggl et al., 2014 [210]	Endurance athletes	Fluid solution; 0.3 g/kg BM	Mild	NO	Improvements in lactate, blood pH, and HCO_3_^−^ were found while supplementing NaHCO_3_ (*p* < 0.01)
Christensen et al., 2014 [211]	Lightweight rowers	Capsule; 0.3 g/kg BM	None	YES/NO	Solely, NaHCO_3_ has no effect, but combined with caffeine (ES: 0.6; *p* < 0.01)
Egger et al., 2014 [212]	Trained cyclists	Fluid solution; 0.3 g/kg BM	None	YES	Cycling time to exhaustion was improved under NaHCO_3_ compared to a placebo (ES: 0.6; *p* < 0.05)
Thomas et al., 2016 [213]	Trained cyclists	Capsule; 0.3 g/kg BM	N/A	YES	Lesser VO_2_ and VE decrease during trial while supplementing NaHCO_3_ (r = 0.74; *p* < 0.01)
Freis et al., 2017 [214]	Endurance ath- letes	Fluid Solution; 0.3 g/kg BM	Severe	YES/NO	NaHCO_3_ led to no change in time to exhaustion but higher maximum running speed (*p* = 0.009)

g: gram; kg: kilogram; BM: body mass; min: NaHCO_3_: sodium bicarbonate; BA: beta-alanine; N/A: not applicable; Mild: minimum discomfort; None: no discomfort; VO2: oxygen consumption; VE: ventilation; F: Fishers F test; d = Cohen’s d; ES: effect size; r = Pearson correlation; Severe: serious discomfort; η^2^ = eta squared. * The study examined the effects of both short-term and long-term use of the substance, and these were analyzed independently.

**Table 11 nutrients-16-03949-t011:** Summary of research on the impact of sodium bicarbonate on performance during physical activity.

	*n*	ST	SS	Stroke		
				L	R	I
Gao et al., 1988 [215]	10 ♂		+			
Pierce et al., 1992 [216]	7 ♂	?				
Zajac et al., 2009 [199]	8 ♂	−	+			
Lindh et al., 2008 [197]	9 ♂	−				
Siegler et al., 2010 [200]	8 ♀ and 6 ♂	−				
Campos et al., 2012 [217]	3 ♀ and 7♂	?		?	?	?
Joyce et al., 2012 [202]	8 ♂	?				
Kumstát et al., 2018 [218]	6 ♂	?				
Yong et al., 2018 [195]	8 ♂	−				

ST: swimming time; SS: swimming speed; L: length; R: rate; I: index; Green colour: positive impact on performance; yellow colour: undetermined impact on performance; red colour: negative impact on performance.

**Table 12 nutrients-16-03949-t012:** Summary of studies investigating the impact of sodium bicarbonate, when combined with beta-alanine, caffeine, creatine, and nitrates, on enhancing exercise performance.

	*n*	Swimming Time			
		SB		SB, BA, SB + BA	SB + Cr
Mero et al., 2004 [226]	8 ♀ and 8 ♂				−
Pruscino et al., 2008 [198]	6 ♂	?			
de Salles Painelli et al., 2013 [225]	7 ♀ 7 ♂		−	−	
Mero et al., 2013 [205]	13 ♂		−		

Red colour: no impact; yellow colour: indeterminate impact.

## Data Availability

The data associated with this article are not publicly accessible but can be obtained by contacting the corresponding author.

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
