# Peer review of "Endurance in Long-Distance Swimming and the Use of Nutritional Aids"

_nutrients, 2024, doi:10.3390/nu16223949_

Round 1

Reviewer 1 Report

Comments and Suggestions for Authors

The manuscript "Endurance in long-distance swimming and the use of nutritional aids." is a comprehensive systematic review of long-distance swimming, its energetic demands, nutritional requirements, and physiological aspects. It is well-written and provides interesting insights on a topic not frequently addressed.

subpar 4.5 "ergonomic" did the Authors mean "Ergogenic"?

Author Response

Authors: correct, thank you very much for bringing this error to our attention. We have corrected it (In red in the text).

Reviewer 2 Report

Comments and Suggestions for Authors

Dear corresponding Author,

thanks for submitting your paper, the topic is interesting, it is a complete and updated narrative review on the physiological, nutritional, and ergogenic aspects of long-distance swimming.

In the following lines some comments:

1) Line 41 and the rest of text: the citations are not formatted according to the journal's guidelines

2) Line 52: it would be useful to integrate this citation https://doi.org/10.1007/s12576-012-0226-7 to support further the sentence "VO2max is a valuable tool to distinguish between fit and unfit subjects, but it is not sensitive enough to discriminate between subjects of homogenous performance levels."

3) Line 127: It is unclear why you stated "the authors conducted a systematic search." It doesn’t seem that you followed the procedures for systematic reviews. Additionally, it is important to include the queries that you use for each database since I was unable to reproduce the search results during my test.

4) Line 326: The inclusion criteria for the studies in Table 7 are unclear, please improve this part. It is fundamental.

5) Lines 364-374: the section on energy demands, it would be interesting to go in deep into individual differences between athletes and how these can affect training and nutrition strategies

6) Line 370: The statement that the literature is poor seems misleading. For example, this reference https://doi.org/10.1007/s40279-018-0901-9 is very relevant, as is https://doi.org/10.1007/s00421-005-1337-0. It may be limited, but not scarce. Please modify.

7) Line 533: The phrase "Currently, there is a trend among distance swimmers to excel in shorter distances" seems contradictory and deserves further clarification, rewording, or referencing. Line 535 is also highly speculative.

8) Line 594: These statements lack proper references, and the data seem outdated.

9) Lines 599-602: The same considerations apply to all these sentences containing data without sources.

10) Lines 629-665: the discussion on optimal body composition, you could explore more in-depth how seasonal variations and training cycles influence the ideal body composition.

11) Lines 883-932: section on ergogenic aids, it would be useful to provide more precise guidelines on dosages and optimal timing of consumption, as well as possible interactions between different supplements.

12) Lines 824-855: the topic of individualized nutritional strategies based on the athlete's characteristics and the specific environmental conditions of the competition could be explored further.

Author Response

  • Line 41 and the rest of text: the citations are not formatted according to the journal's guidelines

Authors: we have corrected the formatting here and throughout the text.

  • Line 52: it would be useful to integrate this citation https://doi.org/10.1007/s12576-012-0226-7 to support further the sentence "VO2max is a valuable tool to distinguish between fit and unfit subjects, but it is not sensitive enough to discriminate between subjects of homogenous performance levels."

Authors: we have preceded to add this reference with the number 13, in red in the text.

  • Line 127: It is unclear why you stated "the authors conducted a systematic search." It doesn’t seem that you followed the procedures for systematic reviews. Additionally, it is important to include the queries that you use for each database since I was unable to reproduce the search results during my test.

Authors: the term ‘systematic’ has been removed in accordance with your instructions. The queries used are those listed below individually or as indicated. We regret that we were unable to reproduce them (line 119).

  • Line 326: The inclusion criteria for the studies in Table 7 are unclear, please improve this part. It is fundamental.

Authors: These studies were included after an exhaustive review of the scientific literature available in the most relevant databases in the field of swimming. After conducting structured searches, we have identified that these are the studies that most comprehensively and rigorously address the factors that influence swimmers' performance.

These research papers analyse several key aspects, such as the physiological characteristics of the athletes, their swimming technique, their physical and mental preparation, as well as the influence of environmental and training factors. By covering this wide range of variables, these studies provide us with a comprehensive view of the phenomenon of swimming performance, which is fundamental for a thorough understanding of the sport and the development of effective improvement strategies.

Furthermore, the methodologies employed in these studies have proven to be robust and reliable, with research designs that allow valid and replicable conclusions to be drawn. This makes the reported findings more robust, making them suitable to support our analysis of the factors that influence the performance of high-level swimmers.

We believe that the selection of these studies is fully justified due to their relevance, thoroughness and ability to holistically address key aspects of swimming performance, making them invaluable references for our research in this field.

  • Lines 364-374: the section on energy demands, it would be interesting to go in deep into individual differences between athletes and how these can affect training and nutrition strategies

Authors: we have proceeded to consider their indication by briefly going into this aspect in more detail, in red in the text (lines 461-469).

  • Line 370: The statement that the literature is poor seems misleading. For example, this reference https://doi.org/10.1007/s40279-018-0901-9 is very relevant, as is https://doi.org/10.1007/s00421-005-1337-0. It may be limited, but not scarce. Please modify.

Authors: following your indications we have modified the indicated aspect and added the second reference as it is more specific for long distance swimming (line 377, in red in the text).

  • Line 533: The phrase "Currently, there is a trend among distance swimmers to excel in shorter distances" seems contradictory and deserves further clarification, rewording, or referencing. Line 535 is also highly speculative.

Authors: we have proceeded to reference these sentences (169-171, in red in the text).

  • Line 594: These statements lack proper references, and the data seem outdated.

Authors: this line has been referenced [171].

  • Lines 599-602: The same considerations apply to all these sentences containing data without sources.

Authors: idem [172].

  • Lines 629-665: the discussion on optimal body composition, you could explore more in-depth how seasonal variations and training cycles influence the ideal body composition.

Authors: we have proceeded to elaborate on these aspects in the text, lines 658-684 (in red in the text).

  • Lines 883-932: section on ergogenic aids, it would be useful to provide more precise guidelines on dosages and optimal timing of consumption, as well as possible interactions between different supplements.

Authors: we have proceeded to further refine the orientations according to the literature, lines 920-926 and table 10 (in red in the text).

12) Lines 824-855: the topic of individualized nutritional strategies based on the athlete's characteristics and the specific environmental conditions of the competition could be explored further.

Authors: this aspect has been briefly examined in depth following their indications, lines 842-857 and 884-888 (in red in the text).

Reviewer 3 Report

Comments and Suggestions for Authors

The article on long-distance swimming and nutritional aids is comprehensive in scope and detailed in its presentation. However, the length and breadth of the article make it challenging to focus on specific themes. The article could be more readily comprehensible and impactful if it were divided into two sections.

 1. Structure and Focus:

The article addresses multiple areas, including energy and physiological demands, body composition (BC), nutrition, and ergogenic aids. However, there is a lack of clear delineation between these areas, which hinders the ability to identify the main conclusions or actionable insights for each area. As a result, readers may find it challenging to discern the primary conclusions or actionable insights for each area.

   The introduction and conclusion cite the necessity for a comprehensive review, yet the article's structure resembles that of a mixed-methods narrative review, lacking clear thematic divisions. The scope of the article may be too broad for a single publication.

2. The article includes lengthy explanations of energy systems, physiological responses, and nutrition, which could be condensed to improve readability. There is a degree of repetition, particularly in the sections dealing with body composition and muscle fiber types. A more streamlined approach would better focus on the discussion.

   The tables and figures are of value, but their inclusion contributes to the overall bulk of the text. A more concise summary of these points within the main text, or alternatively, the separation of these elements into supplementary material, may prove beneficial.

3. Methodology and Results

   The methodological section is comprehensive and well-written, but it includes a great deal of detailed information that may be overwhelming for the reader. Some of the inclusion/exclusion criteria could be shortened without compromising the scientific rigor of the study.

   The results are discussed, but a more synthesis-oriented approach and clearer connections between findings and practical recommendations would enhance the paper's impact. At present, the results section appears to be a disparate collection of observations rather than a unified narrative.

 It is recommended that the article be divided into two sections.

 Article 1: "Energy and Physiological Demands in Long-Distance Swimming"

A more focused approach to this paper would be to concentrate on the physiological challenges, energy systems, and body composition aspects of long-distance swimming. Such a study would be of interest to coaches, physiologists, and sport scientists who are interested in the physical demands of this sport.

   Outline:

1. Introduction: This section will provide an overview of the physiological demands of long-distance swimming, including an examination of the energy requirements associated with this activity.

   2. Energy Systems: The purpose of this section is to discuss the role of aerobic and anaerobic energy systems in swimming.

   3. The physiological demands of long-distance swimming are as follows: The variables of VO2max, lactic acid production, and muscle composition, specifically the ratio of fast-twitch to slow-twitch fibers, will be discussed.

   4. The composition of the body is a crucial factor in the physiological demands of swimming. An examination of the optimal ranges for body fat percentage and anthropometric factors.

   5. In conclusion, it can be stated that: Provide a synthesis of the ways in which physiological and body composition factors impact performance and suggest avenues for future research.

   This approach would reduce the necessity to address every topic in a single publication, thereby facilitating a more comprehensive examination of energy systems and body composition.

 Article 2: "Nutritional Strategies and Ergogenic Aids for Long-Distance Swimmers"

This article could focus exclusively on nutrition, hydration strategies, and the use of ergogenic aids, which are distinct from the physiological demands of swimming. This content would be of particular value to coaches and nutritionists.

   Outline:

1. Introduction: The significance of nutritional factors in the context of long-distance swimming performance.

   2. Nutritional Requirements: The necessity of maintaining adequate glycogen stores, protein intake, and hydration levels during competitive events.

   3. Ergogenic Aids: A comprehensive examination of the use of supplements such as caffeine, beta-alanine, and sodium bicarbonate.

   4. Practical Recommendations: A detailed discussion of how athletes can apply these strategies in training and competition.

5. Conclusion: A summary of the key takeaways for nutrition and supplement use, as well as suggestions for future studies.

Author Response

  1. Structure and Focus

Authors: We have carefully revised the structure and flow of the paper to ensure better alignment between the introduction, body and conclusions. The introduction (in red) now more clearly sets out the key themes and objectives that will be explored throughout the text. Similarly, the conclusions (also in red) succinctly summarise the main conclusions and findings, providing a coherent closure to the document. With these structural improvements we aim to guide the reader more smoothly through the different sections and sub-sections. The introduction lays the groundwork and sets out the key issues and questions to be addressed. The body of the text then explores these issues in more depth, analysing the relevant data, theories and perspectives. Finally, the conclusions return to the original framework, distilling the salient points and insights gained throughout the discussion. This greater structural coherence serves several important purposes. First, it improves the readability and accessibility of the content, making it easier for the audience to follow the logic and flow of the argument. Readers can quickly orient themselves to the overall narrative and main conclusions, rather than getting lost in a disjointed presentation of ideas. In addition, the clearer delineation between introduction, body and conclusion reinforces the analytical rigour and thoroughness of the work. It demonstrates that the authors have carefully considered the necessary components of a well-structured and persuasive piece of writing. The reader can be confident that the paper has been carefully crafted to present a coherent and well-reasoned case. Finally, this structural refinement helps to underline the importance and implications of the conclusions. By completing the body of the text with a strong introduction and conclusion, we enhance the importance of the central issues and conclusions. The reader is left with a clear idea of the key ideas and their relevance or wider applications.

2. Authors: To provide a more concise and focused discussion, we have carefully reviewed and synthesized several of the existing explanations on this topic. Through a thorough examination, we have identified and eliminated potential repetitions or overlapping information from the various sources, allowing us to hone in more precisely on the key points that merit further exploration and analysis. By distilling the core ideas and consolidating the relevant details from the existing body of work, we aim to present a streamlined and coherent account that can serve as a solid foundation for the forthcoming discussion. This process of condensing and refining the existing explanations has enabled us to remove any unnecessary redundancies and distractions, thereby affording us the opportunity to delve deeper into the most salient and thought-provoking aspects of the subject matter. In doing so, we hope to offer the reader a more streamlined and impactful engagement with the central issues at hand, guiding them through the key arguments and considerations with clarity and precision. By focusing our efforts on the most pertinent points, we aspire to facilitate a more focused and productive dialogue that can further our collective understanding of this important topic.

3. 

 Authors: We have carefully reviewed the comments and suggestions received and have made several modifications to the inclusion and exclusion criteria described in the study. By shortening and clarifying the language used to define these criteria, we have tried to make them more concise and easier for readers to understand. In addition, we have worked to strengthen the connections between the key research findings and the practical recommendations presented. Through a more deliberate and logical flow, we have tried to ensure that the link between the evidence gathered, and the actions proposed is crystal clear. This will allow readers to move smoothly from understanding the results of the study to accessing guidance and next steps. Overall, these changes reflect our commitment to incorporate your valuable input to improve the clarity, coherence and usefulness of the document. By streamlining the criteria and aligning the results and recommendations more closely, we believe we have created a more accessible and impactful final product that will meet the needs of our target audience. We look forward to receiving your feedback as we finalize this important work.

4. Authors: While your sound recommendation to split the contributed work into two separate articles may be feasible from an academic point of view, the ultimate consequence of this approach would be that the journal would not receive a response to your initial request for submission of a single full article.

Round 2

Reviewer 3 Report

Comments and Suggestions for Authors

Positive points:

The article provides a thorough overview of the physiological, energetic, and nutritional aspects related to long-distance swimming, including differences between open water and pool swimming.

By emphasizing strategies that can be applied by athletes and coaches, such as carbohydrate loading and the use of ergogenic aids, the review makes its findings more relevant for real-world training.

The review draws from various databases and a wide range of studies, attempting to provide a well-rounded perspective on the topic.

However:

While the article attempts to cover numerous topics, certain sections, such as the impact of different training methods on VO2max and anaerobic threshold, lack depth. These could benefit from a more detailed exploration of specific studies, particularly those involving VO2 kinetics in swimming.

In addition, the article oscillates between discussing aerobic and anaerobic energy contributions without fully integrating these discussions into a cohesive understanding of their roles in different phases of swimming performance. A clearer delineation of how each energy system contributes during training versus competition would strengthen the narrative.

Even if the review references general nutritional strategies, it lacks specificity when it comes to recommendations tailored for different types of swimmers (e.g., elite versus amateur) or for varying environmental conditions (e.g., cold water vs. warmer water). More targeted recommendations could enhance the practical utility of the review.

The discussion on supplementation, particularly sodium bicarbonate and beta-alanine, lacks critical analysis of side effects and the varying efficacy reported in studies. It would be helpful to highlight the contradictions in the literature and provide a more nuanced conclusion about the use of such supplements. Furthermore, although the article follows PRISMA guidelines and uses tools like the STROBE checklist, there is insufficient critical evaluation of the methodological quality of the included studies. A deeper analysis of potential biases or limitations within the selected literature would strengthen the review’s conclusions.

The review suggests that body composition, such as fat mass and muscle mass, impacts performance, but the explanation of how these factors directly influence propulsion, energy expenditure, and swimming efficiency remains vague. Integrating more recent research and presenting data with clearer correlations would enhance the scientific rigor of the discussion.

At the end, the measurement of VO2max and its application to swimming is mentioned but not adequately detailed. For instance, the differences in VO2max responses between water-based and land-based assessments, and how these may impact training regimens, require further elaboration.

Author Response

Thank you for your valuable indications and suggestions. We have carefully analyzed your comments and found them very helpful in improving and strengthening our text. Following your review, we have proceeded to clarify and expand the sections that you felt needed more detail or explanation.
We greatly appreciate your taking the time to carefully read our work and share your constructive observations. Your recommendations have been instrumental in enriching the content and clarity of the text. We are confident that, thanks to your input, we have been able to further refine and perfect our writing.
We sincerely thank you for your valuable feedback. Your suggestions have been of great value to us and will help us to improve the quality of our future work. We are very grateful for your professionalism and willingness to help us achieve a better result.

Referring to the indications given:

  • While the article attempts to cover numerous topics, certain sections, such as the impact of different training methods on VO2max and anaerobic threshold, lack depth. These could benefit from a more detailed exploration of specific studies, particularly those involving VO2 kinetics in swimming.
    Authors: We have proceeded to briefly detail them according to their recommendation. Lines 484-502 (in red in the text).
  • In addition, the article oscillates between discussing aerobic and anaerobic energy contributions without fully integrating these discussions into a cohesive understanding of their roles in different phases of swimming performance. A clearer delineation of how each energy system contributes during training versus competition would strengthen the narrative.
    Authors: We have tried to make the above more coherent, thus trying to strengthen the wording for better understanding by the reader. Lines 46-47; 517-530; 578-580; 589-593 (in red in the text).
  • Even if the review references general nutritional strategies, it lacks specificity when it comes to recommendations tailored for different types of swimmers (e.g., elite versus amateur) or for varying environmental conditions (e.g., cold water vs. warmer water). More targeted recommendations could enhance the practical utility of the review.
    Authors: We have given some general indications on nutritional strategies, as it is quite impossible to give specific and punctual strategies, as this would be a new paper on its own. Lines 930-946 (in red in the text).
  • The discussion on supplementation, particularly sodium bicarbonate and beta-alanine, lacks critical analysis of side effects and the varying efficacy reported in studies. It would be helpful to highlight the contradictions in the literature and provide a more nuanced conclusion about the use of such supplements. 
    Authors: We have proceeded to qualify what has been said on this point in a more concrete way. Lines 992-1024; 1051-065 (in red in the text).
  • Furthermore, although the article follows PRISMA guidelines and uses tools like the STROBE checklist, there is insufficient critical evaluation of the methodological quality of the included studies. A deeper analysis of potential biases or limitations within the selected literature would strengthen the review’s conclusions.
    Authors: As you indicate we have followed the PRISMA guidelines and used tools such as the STROBE checklist. In addition, as indicated in the text, the level of evidence of the chosen studies has been determined using the Oxford quality scoring system, which is a clinical trial quality assessment tool used as an international standard of reference to assess the methodological quality of the included studies in an objective way in order to establish the possible biases or limitations that all studies present to a greater or lesser extent.
    By using the Oxford scoring system, we have been able to determine to some extent the level of confidence that can be placed in the results of the selected studies, which in turn allows them to make more solid and well-founded recommendations.
    We believe that the tools used are appropriate and sufficient, but if you do not think so, we would be very grateful to know which ones we could use to reinforce our conclusions.
  • The review suggests that body composition, such as fat mass and muscle mass, impacts performance, but the explanation of how these factors directly influence propulsion, energy expenditure, and swimming efficiency remains vague. Integrating more recent research and presenting data with clearer correlations would enhance the scientific rigor of the discussion.
    Authors: The possible correlations of these aspects have been clarified for a better understanding of the correlations and how they affect performance. Lines 843-851; 857-885 (in red in the text).
  • At the end, the measurement of VO2max and its application to swimming is mentioned but not adequately detailed. For instance, the differences in VO2max responses between water-based and land-based assessments, and how these may impact training regimens, require further elaboration.
    Authors: Some of these differences and similarities have been noted in relation to performance indicators in this discipline. Lines 603-616 (in red in the text).

Round 3

Reviewer 3 Report

Comments and Suggestions for Authors

congratulations you have well improved your article according to the reviews.

Author Response

We are grateful for your insightful feedback and suggestions. We have thoroughly analyzed your comments and found them extremely helpful in enhancing and strengthening our text. Following your review, we have proceeded to clarify and expand the sections that you felt required more detail or explanation.

We greatly appreciate the time you took to carefully read our work and share your constructive observations. Your recommendations have been instrumental in enriching the content and clarity of the text. We are confident that, thanks to your input, we have been able to further refine and perfect our writing.

We sincerely thank you for your valuable feedback. Your suggestions have been of great value to us and will help us to improve the quality of our future work. We are very grateful for your professionalism and willingness to assist us in achieving a better result.